# On the Convergence of Projected Alternating Maximization for Equitable and Optimal Transport

## Abstract

This paper studies the equitable and optimal transport (EOT) problem, which has many applications such as fair division problems and optimal transport with multiple agents etc. In the discrete distributions case, the EOT problem can be formulated as a linear program (LP). Since this LP is prohibitively large for general LP solvers, Scetbon *et al.* Scetbon et al. (2021) suggests to perturb the problem by adding an entropy regularization. They proposed a projected alternating maximization algorithm (PAM) to solve the dual of the entropy regularized EOT. In this paper, we provide the first convergence analysis of PAM. A novel rounding procedure is proposed to help construct the primal solution for the original EOT problem. We also propose a variant of PAM by incorporating the extrapolation technique that can numerically improve the performance of PAM. Results in this paper may shed lights on block coordinate (gradient) descent methods for general optimization problems.

## 1 Introduction

Optimal transport (OT) is a classical problem that recently finds many emerging applications in machine learning and artificial intelligence, including generative models Arjovsky et al. (2017), representation learning Ozair et al. (2019), reinforcement learning Bellemare et al. (2017) and word embeddings Alvarez-Melis et al. (2019) etc. More recently, Scetbon *et al.* Scetbon et al. (2021) proposed an equitable and optimal transport (EOT) problem that targets to fairly distribute the workload of OT when there are multiple agents. In this problem formulation, there are multiple agents working together to move mass from measures $\mu$ to $\nu$ and each agent has its unique cost function. A very important issue that needs to be considered here is the fairness, which aims at finding transportation plans such that the workloads among all the agents are equal to each other. This can be achieved by minimizing the largest transportation cost among all agents, which leads to a convex-concave saddle point problem. The EOT problem has wide applications in economics and machine learning, such as fair division or the cake-cutting problem Moulin (2003); Brandt et al. (2016), multi-type resource allocation Mackin & Xia (2015), internet minimal transportation time and sequential optimal transport Scetbon et al. (2021).

We now describe the EOT problem formally. Given two discrete probability measures $\mu_n = \sum_{i=1}^n a_i \delta_{x_i}$ and $\nu_n = \sum_{i=1}^n b_i \delta_{y_i}$, the EOT studies the problem of transporting mass from $\mu$ to $\nu$ by $N$ agents. Here, $\{x_1, x_2, ..., x_n\} \subseteq \mathbb{R}^d$ and $\{y_1, y_2, ..., y_n\} \subset \mathbb{R}^d$ are the support points of each measure and $a = [a_1, a_2, ..., a_n]^\top \in \Delta^n$, $b = [b_1, b_2, ..., b_n]^\top \in \Delta^n$ are corresponding weights for each measure, where $\Delta^n$ denotes the probability simplex in $\mathbb{R}^n$. Moreover, throughout this paper, we assume vector $b > 0$. For each agent $k$, we denote its unique cost function as $c^k(x, y), k \in [N] = \{1, \ldots, N\}$ and its cost matrix as $C^k$, where $C_{i,j}^k = c^k(x_i, y_j)$. Moreover, we define the following coupling decomposition set

$$\Pi_{a,b}^N := \left\{ \boldsymbol{\pi} = (\pi^k)_{k \in [N]} \,\middle|\, r\left(\sum_k \pi^k\right) = a, \quad c\left(\sum_k \pi^k\right) = b, \quad \pi_{ij}^k \geq 0, \forall i, j \in [n] \right\},$$

where $r(\pi) = \pi\mathbf{1}, c(\pi) = \pi^\top\mathbf{1}$ are the row sum and column sum of matrix $\pi$ respectively. Mathematically, the EOT problem can be formulated as

$$\min_{\boldsymbol{\pi}\in\Pi_{a,b}^N} \max_{1\le k\le N} \langle\pi^k, C^k\rangle. \tag{1}$$

When $N = 1$, (1) reduces to the standard OT problem. Note that (1) minimizes the point-wise maximum of a finite collection of functions. It is easy to see that (1) is equivalent to the following constrained problem:

$$\min_{\boldsymbol{\pi}\in\Pi_{a,b}^N} \max_{\lambda\in\Delta_+^N} \ell(\boldsymbol{\pi}, \lambda) := \sum_{k=1}^N \lambda_k\langle\pi^k, C^k\rangle. \tag{2}$$

The following proposition shows an important property of EOT: at the optimum of the minimax EOT formulation (2), the transportation costs of the agents are equal to each other.

**Proposition 1** *(Scetbon et al., 2021, Proposition 1) Assume that all cost matrices $C^k, k\in[N]$ have the same sign. Let $\boldsymbol{\pi}^* \in \Pi_{a,b}^N$ be the optimal solution of (2). It holds that*

$$\langle(\pi^*)^i, C^i\rangle = \langle(\pi^*)^j, C^j\rangle, \quad \forall i, j \in [N]. \tag{3}$$

Note that Proposition 1 requires all cost matrices to have the same sign. When the cost matrices are all non-negative, (2) solves the transportation problem with multiple agents. When the cost matrices are all non-positive, the cost matrices are interpreted as the utility functions and (2) solves the fair division problem Moulin (2003).

The discrete OT is a linear programming (LP) problem (in fact, an assignment problem) with a complexity of $O(n^3\log n)$ Tarjan (1997). Due to this cubic dependence on the dimension $n$, it is challenging to solve large-scale OT in practice. A widely adopted compromise is to add an entropy regularizer to the OT problem Cuturi (2013). The resulting problem is strongly convex and smooth, and its dual problem can be efficiently solved by the celebrated Sinkhorn's algorithm Sinkhorn & Knopp (1967); Cuturi (2013). This strategy is now widely used in the OT community due to its computational advantages as well as improved sample complexity Genevay et al. (2019). Similar ideas were also used for computing the Wasserstein barycenter Benamou et al. (2015), projection robust Wasserstein distance Paty & Cuturi (2019); Lin et al. (2020); Huang et al. (2021a), projection robust Wasserstein barycenter Huang et al. (2021b). Motivated by these previous works, Scetbon *et al.* Scetbon et al. (2021) proposed to add an entropy regularizer to (2), and designed a projected alternating maximization algorithm (PAM) to solve its dual problem. However, the convergence of PAM has not been studied. Scetbon *et al.* Scetbon et al. (2021) also proposed an accelerated projected gradient ascent algorithm (APGA) for solving a different form of the dual problem of the entropy regularized EOT. Since the objective function of this new dual form has Lipschitz continuous gradient, APGA is essentially the Nesterov's accelerated gradient method and thus its convergence rate is known. However, numerical experiments conducted in Scetbon et al. (2021) indicate that APGA performs worse than PAM. We will discuss the reasons in details later.

**Our Contributions.** There are mainly three issues with the PAM and APGA algorithms in Scetbon et al. (2021), and we will address all of them in this paper. Our results may shed lights on designing new block coordinate descent algorithms. Our main contributions are given below.

(i) The PAM algorithm in Scetbon et al. (2021) only returns the dual variables. How to find the primal solution of (2), i.e., the optimal transport plans $\boldsymbol{\pi}$, was not discussed in Scetbon et al. (2021). In this paper, we propose a novel rounding procedure to find the primal solution. Our rounding procedure is different from the one widely used in the literature Altschuler et al. (2017).

(ii) We provide the first convergence analysis of the PAM algorithm, and analyze its iteration complexity for finding an $\epsilon$-optimal solution to the EOT problem (2). In particular, we show that it takes at most $O(Nn^2\epsilon^{-2})$ arithmetic operations to find an $\epsilon$-optimal solution to (2). This matches the rate of the Sinkhorn's algorithm for computing the Wasserstein distance Dvurechensky et al. (2018).

(iii) We propose a variant of PAM that incorporates the extrapolation technique as used in Nesterov's accelerated gradient method. We name this variant as PAM with Extrapolation (PAME). The iteration complexity of PAME is also analyzed. Though we are not able to prove a better complexity over PAM at this moment, we find that PAME performs much better than PAM numerically.

**Notation.** For vectors $a$ and $b$ with the same dimension, $a./b$ denotes their entry-wise division. We denote $c_\infty := \max_k \|C^k\|_\infty$. Throughout this paper, we assume vector $b > 0$, and we denote $\iota := \min_j \log(b_j)$. We use $\mathbf{1}_n$ to denote the $n$-dimensional vector whose entries are all equal to one. We use $\mathbb{I}_{\mathcal{X}}(x)$ to denote the indicator function of set $\mathcal{X}$, i.e., $\mathbb{I}_{\mathcal{X}}(x) = 0$ if $x \in \mathcal{X}$, and $\mathbb{I}_{\mathcal{X}}(x) = \infty$ otherwise. We denote $c^t = c(\sum_{k=1}^N \pi^k(f^{t+1}, g^t, \lambda^t))$. For integer $N > 0$, we denote $[N] := \{1, \ldots, N\}$. We also denote $\boldsymbol{\pi}(f, g, \lambda) = [\pi^k(f, g, \lambda)]_{k \in [N]}$.

## 2 Projected Alternating Maximization Algorithm

The PAM algorithm proposed in Scetbon et al. (2021) aims to solve the entropy regularized EOT problem, which is given by

$$\min_{\boldsymbol{\pi} \in \Pi_{a,b}^N} \max_{\lambda \in \Delta_+^N} \ell_\eta(\boldsymbol{\pi}, \lambda) := \sum_{k=1}^N p_\eta^k(\pi^k, \lambda) \tag{4}$$

where $\eta > 0$ is a regularization parameter, $p_\eta^k(\pi^k, \lambda) := \lambda_k \langle \pi^k, C^k \rangle - \eta H(\pi^k)$, and the entropy function $H$ is defined as $H(\pi) = -\sum_{i,j} \pi_{i,j}(\log \pi_{i,j} - 1)$. The entropy regularization was first introduced into the OT problem by Cuturi (2013) and is now widely used in the OT community. By adding an entropy regularizer, the primal problem becomes strongly convex and the dual problem is unconstrained and is suitable for alternating maximization. This leads to the Sinkhorn's algorithm which has low per-iteration complexity and thus is scalable. The PAM algorithm proposed by Scetbon et al. (2021) used the same idea for the EOT problem. Note that (4) is a strongly-convex-concave minimax problem whose constraint sets are convex and bounded, and thus the Sion's minimax theorem Sion (1958) guarantees that

$$\min_{\boldsymbol{\pi} \in \Pi_{a,b}^N} \max_{\lambda \in \Delta_+^N} \ell_\eta(\boldsymbol{\pi}, \lambda) = \max_{\lambda \in \Delta_+^N} \min_{\boldsymbol{\pi} \in \Pi_{a,b}^N} \ell_\eta(\boldsymbol{\pi}, \lambda). \tag{5}$$

Now we consider the dual problem of $\min_{\boldsymbol{\pi} \in \Pi_{a,b}^N} \ell_\eta(\boldsymbol{\pi}, \lambda)$. First, we add a redundant constraint $\sum_{k,i,j} \pi_{i,j}^k = 1$ and consider the dual of

$$\min_{\boldsymbol{\pi} \in \Pi_{a,b}^N, \sum_{k,i,j} \pi_{i,j}^k = 1} \ell_\eta(\boldsymbol{\pi}, \lambda). \tag{6}$$

The reason for adding this redundant constraint is to guarantee that the dual objective function is Lipschitz smooth. It is easy to verify that the dual problem of (6) is given by

$$\max_{f,g} \min_{\substack{\sum_{k,i,j} \pi_{i,j}^k = 1, \\ \boldsymbol{\pi} \in (\mathbb{R}_+^{n \times n})^N}} \sum_{k=1}^N \lambda_k \langle \pi^k, C^k \rangle - \eta H(\boldsymbol{\pi}) + f^\top \left( a - r \left( \sum_k \pi^k \right) \right) + g^\top \left( b - c \left( \sum_k (\pi^k)^\top \right) \right), \tag{7}$$

where $f$ and $g$ are the dual variables and $H(\boldsymbol{\pi}) = \sum_k H(\pi^k)$. It is noted that problem (7) admits the following solution:

$$\pi^k(f, g, \lambda) = \frac{\zeta^k(f, g, \lambda)}{\sum_k \|\zeta^k(f, g, \lambda)\|_1}, \quad \forall k \in [N], \tag{8}$$

where

$$\zeta^k(f, g, \lambda) = \exp \left( \frac{f \mathbf{1}_n^\top + \mathbf{1}_n g^\top - \lambda_k C^k}{\eta} \right), \quad \forall k \in [N]. \tag{9}$$

By plugging (8) into (7), we obtain the following dual problem of (6):

$$\max_{f \in \mathbb{R}^n, \, g \in \mathbb{R}^n} \langle f, a \rangle + \langle g, b \rangle - \eta \log \left( \sum_{k=1}^N \|\zeta^k(f, g, \lambda)\|_1 \right) - \eta. \tag{10}$$

Plugging (10) into (5), we know that the entropy regularized EOT problem (4) is equavalent to a pure maximization problem:

$$\max_{f \in \mathbb{R}^n, \, g \in \mathbb{R}^n, \, \lambda \in \Delta^N} F(f, g, \lambda) := \langle f, a \rangle + \langle g, b \rangle - \eta \log \left( \sum_{k=1}^N \|\zeta^k(f, g, \lambda)\|_1 \right) - \eta. \tag{11}$$

Function $F(f, g, \lambda)$ is a smooth concave function with three block variables $(f, g, \lambda)$. We use $(f^*, g^*, \lambda^*)$ to denote an optimal solution of (11), and we denote $F^* = F(f^*, g^*, \lambda^*)$. The PAM algorithm proposed in Scetbon et al. (2021) is essentially a block coordinate descent (BCD) algorithm for solving (11). More specifically, the PAM updates the three block variables by the following scheme:

$$f^{t+1} \in \underset{f}{\operatorname{argmax}}\, F(f, g^t, \lambda^t), \tag{12a}$$

$$g^{t+1} \in \underset{g}{\operatorname{argmax}}\, F(f^{t+1}, g, \lambda^t), \tag{12b}$$

$$\lambda^{t+1} := \operatorname{Proj}_{\Delta^N}\left(\lambda^t + \tau \nabla_\lambda F(f^{t+1}, g^{t+1}, \lambda^t)\right). \tag{12c}$$

Each iteration of PAM consists of two exact maximization steps followed by one projected gradient step. Importantly, the two exact maximization problems (12a)-(12b) have numerous optimal solutions, and we choose to use the following ones:

$$f^{t+1} = f^t + \eta \log\left(\frac{a}{r\left(\sum_{k=1}^N \zeta^k(f^t, g^t, \lambda^t)\right)}\right), \tag{13}$$

$$g^{t+1} = g^t + \eta \log\left(\frac{b}{c\left(\sum_{k=1}^N \zeta^k(f^{t+1}, g^t, \lambda^t)\right)}\right). \tag{14}$$

Furthermore, the optimiality conditions of (12a)-(12b) imply that

$$a - \frac{r\left(\sum_{k=1}^N \zeta^k(f^{t+1}, g^t, \lambda^t)\right)}{\sum_k \|\zeta^k(f^{t+1}, g^t, \lambda^t)\|_1} = 0, \quad b - \frac{c\left(\sum_{k=1}^N \zeta^k(f^{t+1}, g^{t+1}, \lambda^t)\right)}{\sum_k \|\zeta^k(f^{t+1}, g^{t+1}, \lambda^t)\|_1} = 0, \quad \forall t. \tag{15}$$

However, we need to point out that the PAM (12) only returns the dual variables $(f^t, g^t, \lambda^t)$. One can compute the primal variable $\boldsymbol{\pi}$ using (8), but it is not necessarily a feasible solution. That is, $\boldsymbol{\pi}$ computed from (8) does not satisfy $\boldsymbol{\pi} \in \Pi_{a,b}^N$. How to obtain an optimal primal solution from the dual variables was not discussed in Scetbon et al. (2021). For the OT problem, i.e., $N = 1$, a rounding procedure for returning a feasible primal solution has been proposed in Altschuler et al. (2017). However, this rounding procedure cannot be applied to the EOT problem directly. In the next section, we propose a new rounding procedure for returning a primal solution based on the dual solution $(f^t, g^t, \lambda^t)$. This new rounding procedure involves a dedicated way to compute the margins.

## 2.1 THE ROUNDING PROCEDURE AND THE MARGINS

Given $a \in \Delta^n$, $b \in \Delta^n$, and $\boldsymbol{\pi} = \{\pi^k\}_{k \in [N]}$ satisfying $r(\sum_k \pi^k) = a$, we construct vectors $a^k, b^k \in \mathbb{R}^n, k \in [N]$ from the procedure

$$(a^k, b^k)_{k \in [N]} = \operatorname{Margins}(\boldsymbol{\pi}, a, b). \tag{16}$$

The details of this procedure is given below. First, we set $a^k = r(\pi^k)$, which immediately implies $\sum_{k=1}^N a^k = a$. We then construct $b^k$ such that the following properties hold (these properties are required in our convergence analysis later):

(i) $b^k \geq 0$;

(ii) $\sum_{k=1}^N b^k = b$;

(iii) $\sum_{i=1}^n a_i^k = \sum_{j=1}^n b_j^k, \quad \forall k \in [N]$;

(iv) For any fixed $j \in [n]$, the quantities $b_j^k - [c(\pi^k)]_j$ have the same sign for all $k \in [N]$. That is, for any $k$ and $k'$, we have

$$(b_j^k - [c(\pi^k)]_j) \cdot (b_j^{k'} - [c(\pi^{k'})]_j) \geq 0, \tag{17}$$

which provides the following identity that is useful in our convergence analysis later:

$$
\sum_{k=1}^{N} \|b^k - c(\pi^k)\|_1 = \sum_{k=1}^{N} \sum_{j=1}^{n} |b_j^k - [c(\pi^k)]_j| = \sum_{j=1}^{n} \left| \sum_{k=1}^{N} (b_j^k - [c(\pi^k)]_j) \right|
$$
$$
= \sum_{j=1}^{n} \left| b_j - \left[ c \left( \sum_{k=1}^{N} \pi^k \right) \right]_j \right| = \left\| b - c \left( \sum_{k=1}^{N} \pi^k \right) \right\|_1. \tag{18}
$$

The procedure on constructing $(b^k)_{k \in [N]}$ satisfying these four properties is provided in Appendix A.

After $(a^k, b^k)_{k \in [N]}$ are constructed from (16) with $\boldsymbol{\pi} = \boldsymbol{\pi}(f^T, g^{T-1}, \lambda^{T-1})$, we adopt the rounding procedure proposed in Altschuler et al. (2017) to output a primal feasible solution $(\hat{\pi}^k)_{k \in [N]}$. The rounding procedure is described in Algorithm 2.

With this new procedure for rounding and computing the margins $a^k, b^k$, we now formally describe our PAM algorithm in Algorithm 1. Note that the algorithm is terminated when the following criteria are met:

$$
\|c^{t-1} - b\|_1 \le \epsilon / (6(6c_\infty - \eta\iota)), \tag{19a}
$$
$$
\|\lambda^t - \lambda^{t-1}\|_2 \le \eta\epsilon / (18c_\infty^2), \tag{19b}
$$
$$
\tilde{F}(f^t, g^{t-1}, \lambda^{t-1}) \le \epsilon / 6, \tag{19c}
$$

where $\tilde{F}(f, g, \lambda)$ is the suboptimality defined as: $\tilde{F}(f, g, \lambda) = F(f^*, g^*, \lambda^*) - F(f, g, \lambda)$.

---

**Algorithm 1** Projected Alternating Maximization Algorithm

---

1: **Input:** Cost matrices $\{C^k\}_{1 \le k \le N}$, vectors $a, b \in \Delta_+^n$ with $b > 0$, accuracy $\epsilon$. $f^0 = g^0 = [1, ..., 1]^\top$, $\lambda^0 = [1/N, ..., 1/N]^\top \in \Delta_+^N$. $t = 0$
2: Choose parameters as
$$
\eta = \min \left\{ \frac{\epsilon}{3(\log(n^2 N) + 1)}, c_\infty \right\}, \quad \tau = \eta / c_\infty^2. \tag{20}
$$

3: **while** (19) is not met **do**
4:     Compute $f^{t+1}$ by (13)
5:     Compute $g^{t+1}$ by (14)
6:     Compute $\lambda^{t+1}$ by (12c)
7:     $t \leftarrow t + 1$
8: **end while**
9: Assume stopping condition (19) is satisfied at the $T$-th iteration. Compute $(a^k, b^k)_{k \in [N]} =$ Margins$(\boldsymbol{\pi}(f^T, g^{T-1}, \lambda^{T-1}), a, b)$ as in Section 2.1.
10: **Output:** $(\hat{\pi}, \hat{\lambda})$ where $\hat{\pi}^k = \text{Round}(\pi^k(f^T, g^{T-1}, \lambda^{T-1}), a^k, b^k), \forall k \in [N]$, $\hat{\lambda} = \lambda^{T-1}$.

---

---

**Algorithm 2** Round$(\pi, a, b)$

---

1: **Input:** $\pi \in \mathbb{R}^{n \times n}$, $a \in \mathbb{R}_+^n$, $b \in \mathbb{R}_+^n$.
2: $X = \text{Diag}(x)$ with $x_i = \frac{a_i}{r(\pi)_i} \wedge 1$
3: $\pi' = X\pi$
4: $Y = \text{Diag}(y)$ with $y_j = \frac{b_j}{c(\pi')_j} \wedge 1$
5: $\pi'' = \pi' Y$
6: $err_a = a - r(\pi'')$, $err_b = b - c(\pi'')$
7: **Output:** $\pi'' + err_a err_b^\top / \|err_a\|_1$.

---

## 2.2 CONNECTIONS WITH BCD AND BCGD METHODS

We now discsuss the connections between PAM and the block coordinate descent (BCD) method and the block coordinate gradient descent (BCGD) method. For the ease of presentation, we now assume that we are dealing with the following general convex optimization problem with $m$ block variables:

$$\min_{x_i \in \mathcal{X}_i, i=1,\ldots,m} J(x_1, x_2, \ldots, x_m), \tag{21}$$

where $\mathcal{X}_i \subset \mathbb{R}^{d_i}$ and $J$ is convex and differentiable. The BCD method for solving (21) iterates as follows:

$$x_i^{t+1} = \operatorname*{argmin}_{x_i \in \mathcal{X}_i} J(x_1^{t+1}, x_2^{t+1}, \ldots, x_{i-1}^{t+1}, x_i, x_{i+1}^t, \ldots, x_m^t), \tag{22}$$

and it assumes that these subproblems are easy to solve. The BCGD method for solving (21) iterates as follows:

$$x_i^{t+1} = \operatorname*{argmin}_{x_i \in \mathcal{X}_i} \langle \nabla_{x_i} J(x_1^{t+1}, x_2^{t+1}, \ldots, x_{i-1}^{t+1}, x_i, x_{i+1}^t, \ldots, x_m^t), x_i - x_i^t \rangle + \frac{1}{2\tau} \|x_i - x_i^t\|_2^2, \tag{23}$$

where $\tau > 0$ is the step size. The PAM (12) is a hybrid of BCD (22) and BCGD (23), in the sense that some block variables are updated by exactly solving a maximization problem (the $f$ and $g$ steps), and some other block variables are updated by taking a gradient step (the $\lambda$ step). Though this hybrid idea has been studied in the literature Hong et al. (2017); Xu & Yin (2013), their convergence analysis requires the blocks corresponding to exact minimization to be strongly convex. However, in our problem (11), the negative of the objective function is merely convex. Hence we need to develop new convergence proofs to analyze the convergence of PAM (Algorithm 1). How to extend our convergence results of PAM (Algorithm 1) to more general settings is a very interesting topic for future study.

## 3 CONVERGENCE ANALYSIS OF PAM

In this section, we analyze the iteration complexity of Algorithm 1 for obtaining an $\epsilon$-optimal solution to the original EOT problem (2). The $\epsilon$-optimal solution to (2) is defined as follows.

**Definition 2 (see, e.g., Nemirovski (2005))** *We call $(\hat{\boldsymbol{\pi}}, \hat{\lambda}) \in \Pi_{a,b}^N \times \Delta^N$ an $\epsilon$-optimal solution to the EOT problem* (2) *if the following inequality holds:*

$$\max_{\lambda \in \Delta^N} \ell(\hat{\boldsymbol{\pi}}, \lambda) - \min_{\boldsymbol{\pi} \in \Pi_{a,b}^N} \ell(\boldsymbol{\pi}, \hat{\lambda}) \leq \epsilon.$$

*Note that the left hand side of the inequality is the duality gap of* (2).

### 3.1 MAIN RESULT

We now present our main theorem, which gives the iteration complexity of PAM such that (55) is satisfied, and as a result of Lemma 15, an $\epsilon$-optimal solution to the original EOT problem (2) is obtained.

**Theorem 3** *Define $\epsilon' = \epsilon/(6c_\infty - \eta\iota)$, and set $T$ as*

$$T = 5 + \frac{36}{\eta\sqrt{\gamma_0}\epsilon'} + \frac{648 c_\infty^2}{\eta\epsilon} + \frac{28}{\eta\gamma_0\epsilon} = O\left(c_\infty^2 \epsilon^{-2}\right), \tag{24}$$

*where $\gamma_0 = \min\left\{\frac{1}{(2c_\infty - \eta\iota)^2}, \frac{1}{9c_\infty^2}\right\}$ is a constant and we know $\gamma_0 = O(c_\infty^{-2})$. The output pair of Algorithm 1 is an $\epsilon$-optimal solution of the EOT problem* (2).

*Proof.* See Appendix B. □

**Remark 4** *Though our complexity result matches the rate of the Sinkhorn's algorithm in terms of the dependence on $\epsilon$, we argue that EOT is a more difficult problem than the entropic regularized OT, and thus our results are promising. First, EOT is a saddle-point problem while entropic regularized*

*OT is a minimization problem. Second, the extra variable $\lambda$ in EOT requires a gradient projection step in the PAM algorithm, which introduces significant difficulty to the analysis of the convergence behavior. While for Sinkhorn's algorithm it is much easier to analyze, because the dual is unconstrained. Third, since there are multiple agents in EOT, it is more difficult to design the rounding procedure to obtain the primal solution. We also note that the dependence of $c_\infty$ in our result and in the result of Sinkhorn's algorithm Dvurechensky et al. (2018) are both $c_\infty^2$.*

## 4 PROJECTED ALTERNATING MAXIMIZATION WITH EXTRAPOLATION

In this section, we discuss how to accelerate the PAM algorithm (Algorithm 1). It can be shown that the gradient of $F$ in (11) is Lipschitz continuous[1]. Therefore, Scetbon *et al.* Scetbon et al. (2021) proposed to adopt Nesterov's accelerated gradient method Nesterov (2004) to solve (11). Their algorithm, named APGA (Accelerated Projected Gradient Ascent algorithm), iterates as follows:

$$(v, w, z)^\top \leftarrow (f^{t-1}, g^{t-1}, \lambda^{t-1})^\top + \frac{t-2}{t+1} \left( (f^{t-1}, g^{t-1}, \lambda^{t-1})^\top - (f^{t-2}, g^{t-2}, \lambda^{t-2})^\top \right) \quad (25a)$$

$$(f^t, g^t)^\top \leftarrow (v, w)^\top + \frac{1}{L} \nabla_{(f,g)} F(v, w, z) \quad (25b)$$

$$(\lambda^t)^\top \leftarrow \mathrm{Proj}_{\Delta^N} \left( z + \frac{1}{L} \nabla_\lambda F(v, w, z) \right), \quad (25c)$$

where $L$ is the Lipschitz constant of $\nabla F$. Note that APGA treats the problem (11) as a generic convex and smooth problem, and does not take advantage of the special structures of (11). In particular, $f$ and $g$ are updated using gradient ascent steps. This is in contrast to PAM in which $f$ and $g$ are obtained by exact maximizations, which is expected to improve the function value of $F$ more significantly. In the following, we will design an accelerated algorithm that utilizes this property. Our method is called PAME (PAM with Extrapolation) and it incorporates the extrapolation technique to the gradient step for updating $\lambda$, and $f$ and $g$ are still updated using exact maximizations. We note that currently we are not able to prove a better complexity for PAME. Our iteration complexity result in Theorem 22 is in the same order as that of PAM, but numerically we have observed great improvement of PAME over PAM. It is an interesting future topic to study other accelerations to PAM that can provably achieve improved complexity.

A typical iteration of our PAME algorithm is given below:

$$f^{t+1} = f^t + \eta \log \left( \frac{a}{r \left( \sum_k \zeta^k(f^t, g^t, \lambda^t) \right)} \right), \quad (26a)$$

$$g^{t+1} = g^t + \eta \log \left( \frac{b}{c \left( \sum_k \zeta^k(f^{t+1}, g^t, \lambda^t) \right)} \right), \quad (26b)$$

$$y^{t+1} = \mathrm{Proj}_{\Delta^N} \left( \lambda^t + (1-\theta)(\lambda^t - \lambda^{t-1}) \right), \quad (26c)$$

$$\lambda^{t+1} = \mathrm{Proj}_{\Delta^N} \left( y^{t+1} + \tau \nabla_\lambda F(f^{t+1}, g^{t+1}, y^{t+1}) \right). \quad (26d)$$

Here $\theta \in (0, 1)$ is a given parameter for the extrapolation step. We see that steps (26a)-(26b) are the same as (13)-(14) and they are solutions to the exact maximizations (12a)-(12b). Steps (26c)-(26c) give extrapolation to the gradient step for $\lambda$, similar to Nesterov's accelerated gradient method. Note that PAME (26) solves the dual entropy-regularized EOT problem (11). We use the same rounding procedure in Section 2.1 to generate a primal solution to the original EOT problem (1). The complete PAME algorithm is described in Algorithm 3. Note that the algorithm is terminated when the following criteria are met:

$$\|c^{t-1} - b\|_1 \le \epsilon/(6(6c_\infty - \eta\iota), \quad (27a)$$

$$\|\lambda^{t-1} - \lambda^{t-2}\|_2 \le \eta\epsilon/(60(1-\theta)c_\infty^2), \quad (27b)$$

$$\|\lambda^t - y^t\|_2 \le \eta\epsilon/(42c_\infty^2), \quad (27c)$$

$$\tilde{F}(f^t, g^{t-1}, \lambda^{t-1}) \le \epsilon/6. \quad (27d)$$

---

[1]In Lemma 8 we proved that $\nabla_\lambda F$ is Lipschitz continuous. The Lipschitz continuity of $\nabla_f F$ and $\nabla_g F$ can be proved similarly.

---

**Algorithm 3** Projected Alternating Maximization with Extrapolation Algorithm

---

1: **Input:** Cost matrices $\{C^k\}_{1 \leq k \leq N}$, accuracy $\epsilon$, $\theta \in (0, 1)$.
2: **Initialization:** $f^0 = g^0 = [1, ..., 1]^\top$, $\lambda^0 = [1/N, ..., 1/N]^\top \in \Delta^N$.
3: Choose parameters as

$$\eta = \min\left\{\frac{\epsilon}{3(\log(n^2 N) + 1)}, c_\infty\right\}, \quad \tau = \frac{\eta}{2c_\infty^2}. \tag{28}$$

4: **while** (27) is not met **do**
5:    Compute $f^{t+1}$ by (26a)
6:    Compute $g^{t+1}$ by (26b)
7:    Compute $y^{t+1}$ by (26c)
8:    Compute $\lambda^{t+1}$ by (26d)
9:    $t \leftarrow t + 1$
10: **end while**
11: Assume the stop condition (27) is satisfied at the $T$-th iteration. Compute $(a^k, b^k)_{k \in [N]} = $ Margins$(\boldsymbol{\pi}(f^T, g^{T-1}, \lambda^{T-1}), a, b)$ as in Section 2.1.
12: **Output:** $(\hat{\pi}, \hat{\lambda})$ where $\hat{\pi}^k = $ Round$(\pi^k(f^T, g^{T-1}, \lambda^{T-1}), a^k, b^k)$, $\forall k \in [N]$, $\hat{\lambda} = \lambda^{T-1}$.

---

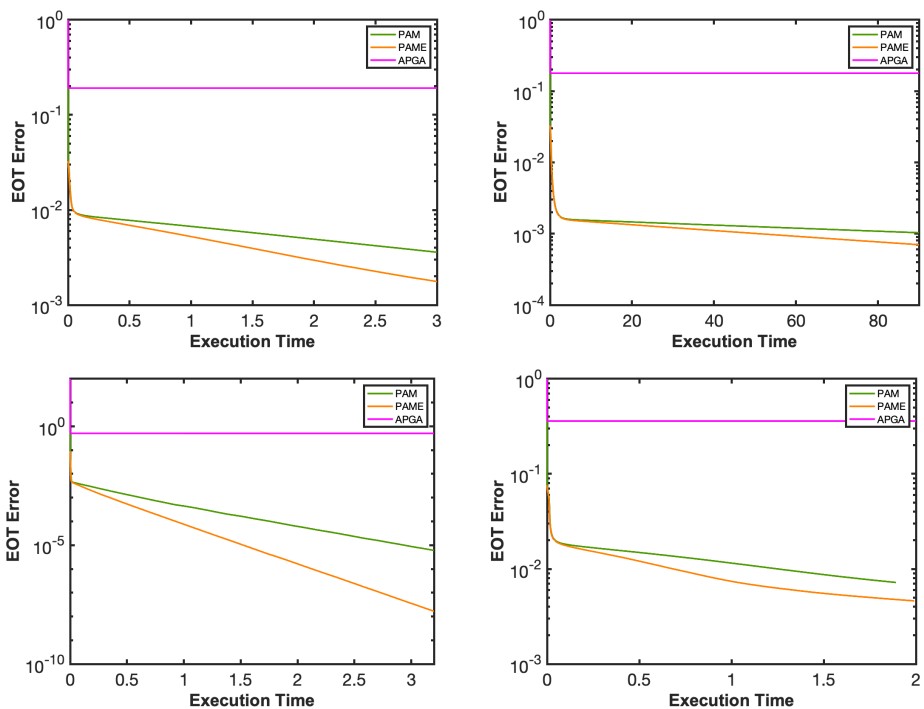

Figure 1: Computational time comparison between PAM, PAME and APGA algorithms on Gaussian distributions. **Upper Left**: $N = 10, n = 100, \eta = 0.1$, **Upper Right**: $N = 10, n = 500, \eta = 0.1$, **Bottom Left**: $N = 10, n = 100, \eta = 0.5$, **Bottom Right**: $N = 5, n = 100, \eta = 0.1$.

## 4.1 CONVERGENCE ANALYSIS OF PAME ALGORITHM

In this section, we analyze the iteration complexity of PAME (Algorithm 3) for obtaining an $\epsilon$-optimal solution to the original EOT problem (1). The proof for PAME is different from that of PAM, and here we need to analyze the behavior of the following Hamiltonian, inspired by Jin *et al.* Jin et al. (2018).

$$E(f, g, \lambda^1, \lambda^2) = F(f, g, \lambda^1) - \frac{1}{2\tau}\|\lambda^1 - \lambda^2\|_2^2. \tag{29}$$

**Theorem 5** *Define $\epsilon' = \epsilon/(6c_\infty - \eta\iota)$, and set $T$ to be*

$$T = 8 + \frac{48}{\eta\sqrt{\gamma_1}\epsilon'} + \frac{\left(3600(1-\theta)^2 + 882\right)c_\infty^2 s}{\eta\epsilon^2} + \frac{48}{\eta\gamma_1\epsilon} = O\left(c_\infty^2 \epsilon^{-2}\right), \tag{30}$$

*where $\gamma_1 = \min\left\{\frac{1}{(2c_\infty - \eta\iota)^2}, \frac{2(2\theta - \theta^2)}{(7-5\theta)^2 c_\infty^2}, \frac{1}{49c_\infty^2}\right\}$ and we know $\gamma_1 = O(c_\infty^{-2})$. At least one of the iterations in Algorithm 3, after rounding, is an $\epsilon$-saddle point of the EOT problem (2).*

*Proof.* See Appendix C. □

**Remark 6** *We are not able to analytically prove that PAME has an improved complexity bound at this moment yet. The APGA proposed in Scetbon et al. (2021) in fact has better complexity than PAM and PAME. However, as demonstrated in Scetbon et al. (2021) and in our numerical experiments (Sections 5 and D), APGA performs worse than PAM. We believe the reason is that APGA takes gradient step for the variables $f$ and $g$, while PAM exactly minimizes the subproblems corresponding to these two variables. It is the exact minimization step that led to the improvement. Developing a provably better algorithm is definitely important and interesting, and we will work on it in the future.*

## 5 NUMERICAL EXPERIMENTS

We compare the performance of PAME with PAM and APGA (25) Scetbon et al. (2021) on a synthetic dataset: the Gaussian distributions. We also conduct numerical comparison on another synthetic dataset: the fragmented hypercube dataset, and the results will be given in the Appendix.

**Gaussian Distribution:** Consider the case when two sets of discrete support $\{x_i\}_{i\in[n]}, \{y_j\}_{j\in[n]}$ are independently sampled from Gaussian distributions

$$\mathcal{N}\left(\left(\begin{array}{c} 1 \\ 1 \end{array}\right), \left(\begin{array}{cc} 10 & 1 \\ 1 & 10 \end{array}\right)\right) \text{ and } \mathcal{N}\left(\left(\begin{array}{c} 2 \\ 2 \end{array}\right), \left(\begin{array}{cc} 1 & -0.2 \\ -0.2 & 1 \end{array}\right)\right) \tag{31}$$

respectively. The base cost matrix $C^{base}$ is computed by $C^{base}_{i,j} = \|x_i - y_j\|_2^2$. Assume we have $N$ agents. The cost matrix of each agent can be obtained by adding Gaussian noise sampled from $\mathcal{N}(0,10)$ to each element of the base cost. For instance, for the $k$-th agent with a cost matrix $C^k$, we have $C^k_{i,j} = |C^{base}_{i,j} + \mathcal{N}(0,10)|$.

We then set $a = b = [1/n, ..., 1/n]$ for all experiments. For all algorithms, we set $\tau = \frac{5\eta}{c_\infty^2}$ and we set $\theta = 0.1$ for the PAME algorithm. We consider the EOT error as a measure of optimality. The EOT error at iteration $t$ is defined by

$$Error = |\ell(\pi(f^t, g^t, \lambda^t), \lambda^t) - \ell^*|, \tag{32}$$

where $\ell^*$ is the approximated optimal value of EOT (2) obtained by running the PAM algorithm for 20000 iterations. Figures 1 plots the EOT error against the execution time for Gaussian distributions. We run each algorithm for 2000 iterations for different parameter settings. In all cases, the PAME and PAM perform significantly better than APGA, and PAME also shows significant improvement over PAM.

## 6 CONCLUSION

In this paper, we provided the first convergence analysis of the PAM algorithm for solving the EOT problem. Specifically, we have shown that it takes at most $O(\epsilon^{-2})$ iterations for the PAM algorithm to find an $\epsilon$-saddle point. We proposed a PAME algorithm which incorporates the extrapolation technique to PAM. The PAME shows significant numerical improvement over PAM. Results in this paper might shed lights on designing new BCD type algorithms.

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

# A  CONSTRUCTING $b^k$ IN THE MARGINS PROCEDURE (16)

In the section, we show how to construct $(b^k)_{k \in [N]}$ in the Margins procedure (16) such that the four properties in Section 2.1 are satisfied.

First, we set

$$b^k = c \left( \pi^k(f^T, g^{T-1}, \lambda^{T-1}) \right) + \frac{b - c \left( \sum_k \pi^k(f^T, g^{T-1}, \lambda^{T-1}) \right)}{N}.$$

It is easy to verify that properties (ii)-(iv) are satisfied. But it is possible that (i) is violated. We now describe a procedure to iteratively update $b^k$ to achieve (i) while keeping (ii)-(iv) satisfied. If (i) does not hold, then there exist $k$ and $j$, such that $b_j^k < 0$, which further implies $b_j^k - [c(\pi^k)]_j < 0$. Since

$$\sum_j b_j^k = \|a^k\|_1, \quad \text{and} \quad \sum_j [c(\pi^k)]_j = \sum_{ij} \pi_{ij}^k = \|a^k\|_1,$$

there must exist an $j'$ such that $b_{j'}^k - [c(\pi^k)]_{j'} > 0$, which further implies $b_{j'}^k > 0$. Moreover, since $\sum_k b_j^k = b_j > 0$, there must also exists an $k'$ such that $b_j^{k'} > 0$. We then update the following quantities:

$$b_j^k \leftarrow b_j^k + \theta$$
$$b_{j'}^k \leftarrow b_{j'}^k - \theta$$
$$b_j^{k'} \leftarrow b_j^{k'} - \theta$$
$$b_{j'}^{k'} \leftarrow b_{j'}^{k'} + \theta,$$

where

$$\theta = \min\{|b_j^k|, |b_j^{k'}|, |b_{j'}^k - [c(\pi^k)]_{j'}|\}.$$

Note that this update maintains that (ii)-(iv) are satisfied. From our discussion above, it is guaranteed that $\theta > 0$. Therefore, $b_j^k$ is improved, i.e., it is getting closer to 0, if not equal. Repeating this procedure leads to $b^k, k \in [N]$ such that (i) is also satisfied.

# B  PROOF OF THEOREM 16

## B.1  TECHNICAL PREPARATIONS

We first give the partial gradients of $F$.

$$[\nabla_f F(f, g, \lambda)]_i = a_i - \frac{\sum_{k,j} \exp((f_i + g_j - \lambda_k C_{ij}^k)/\eta)}{\sum_k \|\zeta^k(f, g, \lambda)\|_1} = a_i - \left[ r \left( \sum_k \pi^k(f, g, \lambda) \right) \right]_i,$$
(33a)

$$[\nabla_g F(f, g, \lambda)]_j = b_j - \frac{\sum_{k,i} \exp((f_i + g_j - \lambda_k C_{ij}^k)/\eta)}{\sum_k \|\zeta^k(f, g, \lambda)\|_1} = b_j - \left[ c \left( \sum_k \pi^k(f, g, \lambda) \right) \right]_j,$$
(33b)

$$[\nabla_\lambda F(f, g, \lambda)]_k = \frac{\sum_{i,j} C_{ij}^k \exp((f_i + g_j - \lambda_k C_{ij}^k)/\eta)}{\sum_k \|\zeta^k(f, g, \lambda)\|_1} = \langle \pi^k(f, g, \lambda), C^k \rangle.$$
(33c)

Since (13) and (14) renormalize the row sum and column sum of $\sum_k \zeta^k(f, g, \lambda)$ to be $a$ and $b$, we immediately have

$$\sum_{k=1}^N \|\zeta^k(f^{t+1}, g^t, \lambda^t)\|_1 = 1, \quad \sum_{k=1}^N \|\zeta^k(f^{t+1}, g^{t+1}, \lambda^t)\|_1 = 1, \forall t,$$
(34)

which, combined with (8), yields

$$\pi^k(f^{t+1}, g^t, \lambda^t) = \zeta^k(f^{t+1}, g^t, \lambda^t), \quad \pi^k(f^{t+1}, g^{t+1}, \lambda^t) = \zeta^k(f^{t+1}, g^{t+1}, \lambda^t), \forall t.$$
(35)

The following lemma gives an error bound for Algorithm 2 (see Altschuler et al. (2017)).

**Lemma 7 (Rounding Error)** *Let* $a, b \in \mathbb{R}_+^n$ *with* $\sum_{i=1}^n a_i = \sum_{j=1}^n b_j = q$, $\pi \in \mathbb{R}_+^{n \times n}$, *and* $\hat{\pi} = Round(\pi, a, b)$. *The following inequality holds:*

$$\|\hat{\pi} - \pi\|_1 \leq 2(\|r(\pi) - a\|_1 + \|c(\pi) - b\|_1).$$

*Proof.* The proof is a slight modification from (Altschuler et al., 2017, Lemma 7). Note that Lines 2-5 in Algorithm 2 renormalize the row sum and column sum that are larger than the corresponding $a_i$ and $b_j$. It is easy to verify that $\hat{\pi}, \pi'', err_a$ and $err_b$ are nonnegative with $\|err_a\|_1 = \|err_b\|_1 = q - \|\pi''\|_1$ and

$$r(\hat{\pi}) = r(\pi'') + r(err_a err_b^\top / \|err_a\|_1) = r(\pi'') + err_a = a,$$

and likewise $c(\hat{\pi}) = b$. Denote $\Delta = \|\pi\|_1 - \|\pi''\|_1$. Since we remove mass from a row of $\pi$ when $r_i(\pi) \geq a_i$, and from a column when $c_j(\pi') \geq b_j$, we have

$$\Delta = \sum_{i=1}^n (r_i(\pi) - a_i)_+ + \sum_{j=1}^n (c_j(\pi') - b_j)_+ .$$

Firstly, a simple calculation shows

$$\sum_{i=1}^n (r_i(\pi) - a_i)_+ = \frac{1}{2} \left[ \|r(\pi) - a\|_1 + \|\pi\|_1 - q \right] .$$

Secondly, the fact that the vector $c(\pi)$ is entrywise larger than $c(\pi')$ leads to

$$\sum_{j=1}^n (c_j(\pi') - b_j)_+ \leq \sum_{j=1}^n (c_j(\pi) - b_j)_+ \leq \|c(\pi) - b\|_1.$$

Therefore we conclude

$$\|\hat{\pi} - \pi\|_1 \leq \Delta + \|err_a err_b^\top\|_1 / \|err_a\|_1 = \Delta + q - \|\pi''\|_1 = 2\Delta + q - \|\pi\|_1$$

$$\leq \|r(\pi) - a\|_1 + 2\|c(\pi) - b\|_1 \leq 2\Big[ \|r(\pi) - a\|_1 + \|c(\pi) - b\|_1 \Big].$$

$\square$

The following lemma shows that $\nabla_\lambda F$ is Lipschitz continuous.

**Lemma 8** *For any* $f, g \in \mathbb{R}^n$ *and* $\lambda^1, \lambda^2 \in \Delta^N$, *the following inequality holds*

$$\|\nabla_\lambda F(f, g, \lambda^1) - \nabla_\lambda F(f, g, \lambda^2)\|_2 \leq c_\infty^2 \|\lambda^1 - \lambda^2\|_2 / \eta, \tag{36}$$

*which immediately implies*

$$F(f, g, \lambda^1) \geq F(f, g, \lambda^2) + \langle \nabla_\lambda F(f, g, \lambda^2), \lambda^1 - \lambda^2 \rangle - \frac{c_\infty^2}{2\eta} \|\lambda^1 - \lambda^2\|_2^2. \tag{37}$$

*Proof.* The proof essentially follows Scetbon et al. (2021). It is easy to verify that the $(q, k)$-th entry of the Hessian of $F(f, g, \lambda)$ with respect to $\lambda$ is

$$\frac{\partial^2 F}{\partial \lambda_q \partial \lambda_k} = \frac{1}{\eta \nu^2} \left[ \sigma_{q,1}(\lambda) \sigma_{k,1}(\lambda) - \nu(\sigma_{k,2}(\lambda) \mathbb{1}_{k=q}) \right]$$

where $\mathbb{1}_{k=q} = 1$ iff $k = q$ and 0 otherwise, for all $k \in \{1, ..., N\}$ and $p \geq 1$

$$\sigma_{k,p}(\lambda) = \sum_{i,j} (C_{i,j}^k)^p \exp\left( \frac{f_i + g_j - \lambda_k C_{i,j}^k}{\eta} \right),$$

$$\nu = \sum_{k=1}^N \sum_{i,j} \exp\left( \frac{f_i + g_j - \lambda_k C_{i,j}^k}{\eta} \right).$$

Let $v \in \mathbb{R}^N$ satisfying $\|v\|_2 = 1$, and by denoting $\nabla_\lambda^2 F$ the Hessian of $F$ with respect to $\lambda$ for fixed $f, g$, we obtain

$$v^\top \nabla_\lambda^2 F v = \frac{1}{\eta \nu^2} \left[ \left( \sum_{k=1}^N v_k \sigma_{k,1}(\lambda) \right)^2 - \nu \sum_{k=1}^N v_k^2 \sigma_{k,2} \right]$$

$$\leq \frac{1}{\eta \nu^2} \left( \sum_{k=1}^N v_k \sigma_{k,1}(\lambda) \right)^2 - \frac{1}{\eta \nu^2} \left( \sum_{k=1}^N |v_k| \sqrt{\sum_{i,j} \exp \left( \frac{f_i + g_j - \lambda_k C_{i,j}^k}{\eta} \right)} \sqrt{\sum_{i,j} (C_{i,j}^k)^2 \exp \left( \frac{f_i + g_j - \lambda_k C_{i,j}^k}{\eta} \right)} \right)^2$$

$$\leq \frac{1}{\eta \nu^2} \left[ \left( \sum_{k=1}^N v_k \sigma_{k,1}(\lambda) \right)^2 - \left( \sum_{k=1}^N |v_k| \sum_{i,j} |C_{i,j}^k| \exp \left( \frac{f_i + g_j - \lambda_k C_{i,j}^k}{\eta} \right) \right)^2 \right] \leq 0,$$

where the last three inequalities come from Cauchy Schwartz inequality. Moreover we have

$$v^T \nabla_\lambda^2 F v = \frac{1}{\eta \nu^2} \left[ \left( \sum_{k=1}^N v_k \sigma_{k,1}(\lambda) \right)^2 - \nu \sum_{k=1}^N v_k^2 \sigma_{k,2} \right] \geq -\frac{\sum_{k=1}^N v_k^2 \sigma_{k,2}}{\eta \nu} \geq -\frac{c_\infty^2}{\eta},$$

which completes the proof. $\qquad \square$

The next lemma gives a bound for $g$.

**Lemma 9** *Let $(f^t, g^t, \lambda^t)$ be the sequence generated by Algorithm 1. For any $t \geq 0$, it holds that*

$$\max_j g_j^t - \min_j g_j^t \leq c_\infty - \eta \iota, \tag{38a}$$

$$\max_j g_j^* - \min_j g_j^* \leq c_\infty - \eta \iota. \tag{38b}$$

*Proof.* We prove (38a) first. When $t = 0$, (38a) holds because of the initialization $g^0$. When $t \geq 1$, from (15) we have

$$\sum_{k=1}^N e^{-\lambda_k^{t-1} C_{ij}^k / \eta} \geq \sum_{k=1}^N e^{-\lambda_k^{t-1} \|C^k\|_\infty / \eta} \geq \sum_{k=1}^N e^{-\|C^k\|_\infty / \eta} \geq N e^{-c_\infty / \eta}, \tag{39}$$

where the second inequality is due to $0 \leq \lambda_k^{t-1} \leq 1$. Combining (39) and (34) we get

$$e^{g_j^t / \eta} \cdot N e^{-c_\infty / \eta} \langle \mathbf{1}, e^{f^t / \eta} \rangle \leq \sum_i e^{g_j^t / \eta} \left( \sum_{k=1}^N e^{-\lambda_k^{t-1} C_{ij}^k / \eta} \right) e^{f_i^t / \eta} = b_j \leq 1,$$

which leads to

$$\max_j g_j^t \leq c_\infty - \eta \log(N \langle \mathbf{1}, e^{f^t / \eta} \rangle). \tag{40}$$

Moreover, note that $e^{-\lambda_k^{t-1} C_{ij}^k / \eta} \leq 1$, therefore $\frac{1}{N} \sum_{k=1}^N e^{-\lambda_k^{t-1} C_{ij}^k / \eta} \leq 1$. This fact leads to:

$$e^{g_j^t / \eta} \cdot \langle \mathbf{1}, e^{f^t / \eta} \rangle \geq \sum_i e^{g_j^t / \eta} \left( \frac{1}{N} \sum_{k=1}^N e^{-\lambda_k^{t-1} C_{ij}^k / \eta} \right) e^{f_i^t / \eta} = \frac{1}{N} b_j,$$

which gives

$$\min_j g_j^t \geq \eta \iota - \eta \log(N \langle \mathbf{1}, e^{f^t / \eta} \rangle). \tag{41}$$

Combining (40) with (41) yields (38a). The bound for $g^*$ (38b) can be obtained similarly, by noting that $\boldsymbol{\pi}^* \in \Pi_{a,b}^N$. We omit the details for brevity. $\qquad \square$

**Lemma 10** *Let $\{f^t, g^t, \lambda^t\}$ be generated by PAM (Algorithm 1). The following equality holds.*

$$\sum_k^N \|\pi^k(f^{t+1}, g^{t+1}, \lambda^t) - \pi^k(f^{t+1}, g^t, \lambda^t)\|_1 = \|c^t - b\|_1, \forall t.$$

*Proof.* By (35), we have

$$\sum_k^N \|\pi^k(f^{t+1}, g^{t+1}, \lambda^t) - \pi^k(f^{t+1}, g^t, \lambda^t)\|_1$$

$$= \sum_k^N \sum_{i,j} |e^{(f_i^{t+1} + g_j^{t+1} - \lambda_k^t C_{i,j}^k)/\eta} - e^{(f_i^{t+1} + g_j^t - \lambda_k^t C_{i,j}^k)/\eta}|$$

$$= \sum_k^N \sum_{i,j} [\pi^k(f^{t+1}, g^t, \lambda^t)]_{i,j} \, |b_j/c_j^t - 1| = \sum_j c_j^t |b_j/c_j^t - 1| = \|c^t - b\|_1.$$

$\square$

## B.2 KEY LEMMAS

In this subsection, we provide a few useful lemmas that will lead to our main theorem on the iteration complexity of PAM (Algorithm 1). These lemmas yield the following results: the function $F$ is monotonically increasing (Lemmas 11), the suboptimality of the dual problem can be upper bounded (Lemma 12-14), and the PAM returns an $\epsilon$-optimal solution under conditions (55) (Lemma 15). In Theorem 16 we will show that these conditions can indeed be satisfied.

**Lemma 11** *[Increase of F] Let $\{f^t, g^t, \lambda^t\}$ be generated by PAM (Algorithm 1). The following inequalities hold:*

$$F(f^{t+1}, g^t, \lambda^t) - F(f^t, g^t, \lambda^t) \geq 0 \tag{42a}$$

$$F(f^{t+1}, g^{t+1}, \lambda^t) - F(f^{t+1}, g^t, \lambda^t) \geq \frac{\eta}{2} \|c^t - b\|_1^2 \tag{42b}$$

$$F(f^{t+1}, g^{t+1}, \lambda^{t+1}) - F(f^{t+1}, g^{t+1}, \lambda^t) \geq c_\infty^2 \|\lambda^{t+1} - \lambda^t\|^2/(2\eta). \tag{42c}$$

*Proof.*

First, (42a) is a direct consequence of (12a).

Next, we prove (42b). We have

$$F(f^{t+1}, g^{t+1}, \lambda^t) - F(f^{t+1}, g^t, \lambda^t)$$

$$= \langle g^{t+1} - g^t, b \rangle - \eta \log \left( \sum_{k=1}^N \|\zeta^k(f^{t+1}, g^{t+1}, \lambda^t)\|_1 \right) + \eta \log \left( \sum_{k=1}^N \|\zeta^k(f^{t+1}, g^t, \lambda^t)\|_1 \right)$$

$$= \langle g^{t+1} - g^t, b \rangle = \eta \sum_{j=1}^n b_j \log(b_j/c_j^t) = \eta \mathcal{K}(b\|c^t) \geq \frac{\eta}{2} \|c^t - b\|_1^2,$$

where $\mathcal{K}(x\|y)$ denotes the KL divergence of $x$ and $y$, the second equality is due to (34), the third equality is due to (14), and the last inequality follows the Pinsker's inequality.

Finally, we prove (42c). From the optimality condition of (12c), we know that there exists

$$h(\lambda^{t+1}) \in \partial \mathbb{I}_{\Delta_N}(\lambda^{t+1}) \tag{43}$$

such that

$$\nabla_\lambda F(f^{t+1}, g^{t+1}, \lambda^t) - \frac{1}{\tau}(\lambda^{t+1} - \lambda^t) - h(\lambda^{t+1}) = 0. \tag{44}$$

From (37) we have

$$F(f^{t+1}, g^{t+1}, \lambda^{t+1}) - F(f^{t+1}, g^{t+1}, \lambda^t) \geq \langle \nabla_\lambda F(f^{t+1}, g^{t+1}, \lambda^t), \lambda^{t+1} - \lambda^t \rangle - \frac{c_\infty^2}{2\eta} \|\lambda^{t+1} - \lambda^t\|^2$$

$$= \langle \frac{1}{\tau}(\lambda^{t+1} - \lambda^t) + h(\lambda^{t+1}), \lambda^{t+1} - \lambda^t \rangle - \frac{c_\infty^2}{2\eta} \|\lambda^{t+1} - \lambda^t\|^2$$

$$\geq \langle \frac{1}{\tau}(\lambda^{t+1} - \lambda^t), \lambda^{t+1} - \lambda^t \rangle - \frac{c_\infty^2}{2\eta} \|\lambda^{t+1} - \lambda^t\|^2$$

$$= c_\infty^2 \|\lambda^{t+1} - \lambda^t\|^2/(2\eta),$$

where the first equality is due to (44), the second inequality is due to (43), and the last equality is due to the definition of $\tau$ in (20).

$\square$

Before we bound the suboptimality gap, we need the following lemma.

**Lemma 12** *Let $\{f^t, g^t, \lambda^t\}$ be generated by PAM (Algorithm 1). For any $\lambda \in \Delta^N$, the following inequality holds:*

$$\langle \lambda - \lambda^t, \nabla_\lambda F(f^{t+1}, g^t, \lambda^t) \rangle \leq 3c_\infty^2 \|\lambda^{t+1} - \lambda^t\|_2 / \eta + c_\infty \left\| c^t - b \right\|_1. \tag{45}$$

*Proof.* The optimality condition of (12c) is given by:

$$\langle \lambda - \lambda^{t+1}, \frac{1}{\tau}(\lambda^{t+1} - \lambda^t) - \nabla_\lambda F(f^{t+1}, g^{t+1}, \lambda^t) \rangle \geq 0, \quad \forall \lambda \in \Delta^N, \tag{46}$$

which implies that

$$\begin{aligned}
&\langle \lambda^{t+1} - \lambda, -\nabla_\lambda F(f^{t+1}, g^{t+1}, \lambda^t) \rangle \\
&\leq \langle \lambda - \lambda^{t+1}, \frac{1}{\tau}(\lambda^{t+1} - \lambda^t) \rangle \leq \frac{1}{\tau}\|\lambda - \lambda^{t+1}\|_2\|\lambda^{t+1} - \lambda^t\|_2 \leq 2c_\infty^2 \|\lambda^{t+1} - \lambda^t\|_2/\eta,
\end{aligned} \tag{47}$$

where the last inequality is due to the fact that the diameter of $\Delta_N$ is bounded by $\sqrt{2} \leq 2$. Moreover, we have

$$\begin{aligned}
&\langle \lambda^t - \lambda, \nabla_\lambda F(f^{t+1}, g^{t+1}, \lambda^t) - \nabla_\lambda F(f^{t+1}, g^t, \lambda^t) \rangle \\
&= \sum_k^N (\lambda_k^t - \lambda_k) \cdot \langle \pi^k(f^{t+1}, g^{t+1}, \lambda^t) - \pi^k(f^{t+1}, g^t, \lambda^t), C^k \rangle \\
&\leq \sum_k^N \|\pi^k(f^{t+1}, g^{t+1}, \lambda^t) - \pi^k(f^{t+1}, g^t, \lambda^t)\|_1 \|C^k\|_\infty \\
&\leq c_\infty \|c^t - b\|_1,
\end{aligned} \tag{48}$$

where the equality is due to (33c), and the last inequality is due to Lemma 10. Finally, we have

$$\begin{aligned}
&\langle \lambda^t - \lambda, -\nabla_\lambda F(f^{t+1}, g^t, \lambda^t) \rangle \\
&= \langle \lambda^t - \lambda^{t+1}, -\nabla_\lambda F(f^{t+1}, g^{t+1}, \lambda^t) \rangle + \langle \lambda^{t+1} - \lambda, -\nabla_\lambda F(f^{t+1}, g^{t+1}, \lambda^t) \rangle + \\
&\quad \langle \lambda^t - \lambda, \nabla_\lambda F(f^{t+1}, g^{t+1}, \lambda^t) - \nabla_\lambda F(f^{t+1}, g^t, \lambda^t) \rangle \\
&\leq \|\lambda^t - \lambda^{t+1}\|_2 \cdot \|\nabla_\lambda F(f^{t+1}, g^{t+1}, \lambda^t)\|_2 + 2c_\infty^2 \|\lambda^{t+1} - \lambda^t\|_2/\eta + c_\infty \|c^t - b\|_1,
\end{aligned} \tag{49}$$

where the first inequality is due to (47) and (48). From (33c) we have $\|\nabla_\lambda F(f^{t+1}, g^{t+1}, \lambda^t)\|_2 \leq c_\infty$, which, combined with (49) and the fact that $\eta \leq c_\infty$, yields the desired result. $\square$

The suboptimality of (11) is defined as: $\tilde{F}(f, g, \lambda) = F(f^*, g^*, \lambda^*) - F(f, g, \lambda)$. Note that $\tilde{F}(f, g, \lambda) \geq 0, \forall f, g, \lambda \in \Delta^N$.

**Lemma 13** *Let $(f^t, g^t, \lambda^t)$ be generated by PAM (Algorithm 1). The following inequality holds:*

$$\tilde{F}(f^{t+1}, g^t, \lambda^t) \leq (2c_\infty - \eta\iota)\|c^t - b\|_1 + 3c_\infty^2 \|\lambda^{t+1} - \lambda^t\|_2/\eta.$$

*Proof.* Denote $u^t = (\max_j g_j^t + \min_j g_j^t)/2, u^* = (\max_j g_j^* + \min_j g_j^*)/2$. From (35) we get

$$\langle \mathbf{1}, c^t - b \rangle = \sum_{i=1}^n a_i - \sum_{j=1}^n b_j = 0,$$

which further implies

$$\begin{aligned}
\langle g^t - g^*, c^t - b \rangle &= \langle (g^t - u^t\mathbf{1}) - (g^* - u^*\mathbf{1}), c^t - b \rangle \\
&\leq (\|g^t - u^t\mathbf{1}\|_\infty + \|g^* - u^*\mathbf{1}\|_\infty) \|c^t - b\|_1 \leq (c_\infty - \eta\iota) \|c^t - b\|_1,
\end{aligned} \tag{50}$$

where the last inequality is due to Lemma 9. Now we set $\lambda = \lambda^*$ in (45), and we obtain

$$\langle \lambda^t - \lambda^*, -\nabla_\lambda F(f^{t+1}, g^t, \lambda^t) \rangle \leq 3c_\infty^2 \|\lambda^{t+1} - \lambda^t\|_2/\eta + c_\infty \|c^t - b\|_1. \qquad (51)$$

Since $F(f, g, \lambda)$ is a concave function, we have

$$F(f^*, g^*, \lambda^*) \leq F(f^{t+1}, g^t, \lambda^t) + \langle \nabla F(f^{t+1}, g^t, \lambda^t), (f^*, g^*, \lambda^*) - (f^{t+1}, g^t, \lambda^t) \rangle,$$

which, combining with (33) yields

$$\begin{aligned}
\tilde{F}(f^{t+1}, g^t, \lambda^t) &= F(f^*, g^*, \lambda^*) - F(f^{t+1}, g^t, \lambda^t) \\
&\leq \langle f^{t+1} - f^*, r(\textstyle\sum_{k=1}^N \pi^k(f^{t+1}, g^t, \lambda^t)) - a \rangle + \langle g^t - g^*, c^t - b \rangle \\
&\quad + \langle \lambda^t - \lambda^*, -\nabla_\lambda F(f^{t+1}, g^t, \lambda^t) \rangle \\
&\leq (2c_\infty - \eta\iota)\|c^t - b\|_1 + 3c_\infty^2 \|\lambda^{t+1} - \lambda^t\|_2/\eta,
\end{aligned}$$

where the last inequality follows from (15), (35), (50) and (51). $\qquad \square$

The next lemma shows that the suboptimality gap $\tilde{F}(f, g, \lambda)$ can be bounded by $O(1/t)$.

**Lemma 14** *Let $(f^t, g^t, \lambda^t)$ be generated by PAM (Algorithm 1). The following inequality holds:*

$$\tilde{F}(f^{t+1}, g^{t+1}, \lambda^{t+1}) \leq \frac{4/(\eta\gamma_0)}{t + 1 + 4/(\eta\gamma_0\tilde{F}(f^0, g^0, \lambda^0))},$$

*where $\gamma_0 = \min\left\{ \frac{1}{(2c_\infty - \eta\iota)^2}, \frac{1}{9c_\infty^2} \right\}$ is a constant.*

*Proof.* Combining (42b) and (42c), we have

$$F(f^{t+1}, g^{t+1}, \lambda^{t+1}) - F(f^{t+1}, g^t, \lambda^t) \geq \frac{\eta}{2} \|c^t - b\|_1^2 + c_\infty^2 \|\lambda^{t+1} - \lambda^t\|_2^2/(2\eta). \qquad (52)$$

Therefore, we have

$$\begin{aligned}
&\tilde{F}(f^{t+1}, g^{t+1}, \lambda^{t+1}) - \tilde{F}(f^{t+1}, g^t, \lambda^t) \\
&\leq -\frac{\eta}{2} \|c^t - b\|_1^2 - c_\infty^2 \|\lambda^{t+1} - \lambda^t\|_2^2/(2\eta) \\
&\leq -\frac{\eta}{2}\gamma_0 \cdot \left( ((2c_\infty - \eta\iota)\|c^t - b\|_1)^2 + (3c_\infty^2\|\lambda^{t+1} - \lambda^t\|_2/\eta)^2 \right) \\
&\leq -\frac{\eta}{4}\gamma_0 \left( (2c_\infty - \eta\iota)\|c^t - b\|_1 + 3c_\infty^2\|\lambda^{t+1} - \lambda^t\|_2/\eta \right)^2 \\
&\leq -\frac{\eta}{4}\gamma_0 \tilde{F}(f^{t+1}, g^t, \lambda^t)^2,
\end{aligned} \qquad (53)$$

where the last inequality is from Lemma 13. Dividing both sides of (53) by $\tilde{F}(f^{t+1}, g^{t+1}, \lambda^{t+1}) \cdot \tilde{F}(f^{t+1}, g^t, \lambda^t)$, we have

$$\begin{aligned}
\frac{1}{\tilde{F}(f^{t+1}, g^{t+1}, \lambda^{t+1})} &\geq \frac{1}{\tilde{F}(f^{t+1}, g^t, \lambda^t)} + \frac{\eta}{4}\gamma_0 \cdot \frac{\tilde{F}(f^{t+1}, g^t, \lambda^t)}{\tilde{F}(f^{t+1}, g^{t+1}, \lambda^{t+1})} \\
&\geq \frac{1}{\tilde{F}(f^{t+1}, g^t, \lambda^t)} + \frac{\eta}{4}\gamma_0 \geq \frac{1}{\tilde{F}(f^t, g^t, \lambda^t)} + \frac{\eta}{4}\gamma_0,
\end{aligned} \qquad (54)$$

where the second inequality is due to (53) and the last inequality is from (42a). Summing (54) from 0 to $t$ leads to

$$\frac{1}{\tilde{F}(f^{t+1}, g^{t+1}, \lambda^{t+1})} \geq \frac{1}{\tilde{F}(f^0, g^0, \lambda^0)} + \frac{\eta(t+1)}{4}\gamma_0,$$

which implies the desired result. $\qquad \square$

The next lemma gives sufficient conditions for the PAM algorithm to return an $\epsilon$-optimal solution to the original EOT problem (2).

**Lemma 15** *Assume PAM terminates at the $T$-iteration, i.e.,*

$$\|c^{T-1} - b\|_1 \leq \epsilon/(6(6c_\infty - \eta\iota)), \tag{55a}$$

$$\left\|\lambda^T - \lambda^{T-1}\right\|_2 \leq \eta\epsilon/(18c_\infty^2), \tag{55b}$$

$$\tilde{F}(f^T, g^{T-1}, \lambda^{T-1}) \leq \epsilon/6. \tag{55c}$$

*Then the output $(\hat{\boldsymbol{\pi}}, \hat{\lambda})$ of PAM (Algorithm 1), i.e., $\hat{\pi}^k = Round(\pi^k(f^T, g^{T-1}, \lambda^{T-1}), a^k, b^k)$, $\forall k \in [N]$, $\hat{\lambda} = \lambda^{T-1}$, is an $\epsilon$-optimal solution of the original EOT problem* (2).

*Proof.* According to Definition 2, it is sufficient to show that the output $(\hat{\boldsymbol{\pi}}, \hat{\lambda}) \in \Pi_{a,b}^N \times \Delta^N$ satisfies the following two inequalities:

$$\max_{\lambda \in \Delta^N} \ell(\hat{\boldsymbol{\pi}}, \lambda) - \ell(\hat{\boldsymbol{\pi}}, \hat{\lambda}) \leq \frac{\epsilon}{2}, \tag{56a}$$

$$\ell(\hat{\boldsymbol{\pi}}, \hat{\lambda}) - \min_{\boldsymbol{\pi} \in \Pi_{a,b}^N} \ell(\boldsymbol{\pi}, \hat{\lambda}) \leq \frac{\epsilon}{2}. \tag{56b}$$

We prove (56a) first. For ease of presentation, we denote $\tilde{\boldsymbol{\pi}} = \boldsymbol{\pi}(f^T, g^{T-1}, \lambda^{T-1})$, $\boldsymbol{\pi}^* = \boldsymbol{\pi}(f^*, g^*, \lambda^*)$. Note that $\hat{\pi}^k = Round(\tilde{\pi}^k, a^k, b^k)$, $\forall k \in [N]$. We also denote

$$\bar{\lambda}(\boldsymbol{\pi}) := \underset{\lambda \in \Delta^N}{\mathrm{argmax}} \left\{ \ell(\boldsymbol{\pi}, \lambda) = \sum_{k=1}^N \lambda_k \langle \pi^k, C^k \rangle \right\}. \tag{57}$$

Note that the term on the left hand side of (56a) can be rewritten as

$$\begin{aligned}
&\ell\left(\hat{\boldsymbol{\pi}}, \bar{\lambda}(\hat{\boldsymbol{\pi}})\right) - \ell\left(\hat{\boldsymbol{\pi}}, \hat{\lambda}\right) \\
&= \underbrace{(\ell(\hat{\boldsymbol{\pi}}, \bar{\lambda}(\hat{\boldsymbol{\pi}})) - \ell(\tilde{\boldsymbol{\pi}}, \bar{\lambda}(\tilde{\boldsymbol{\pi}})))}_{(I)} + \underbrace{([\ell(\tilde{\boldsymbol{\pi}}, \bar{\lambda}(\tilde{\boldsymbol{\pi}})) - \eta H(\tilde{\boldsymbol{\pi}})] - [\ell(\boldsymbol{\pi}^*, \lambda^*) - \eta H(\boldsymbol{\pi}^*)])}_{(II)} \\
&\quad + \underbrace{([\ell(\boldsymbol{\pi}^*, \lambda^*) - \eta H(\boldsymbol{\pi}^*)] - [\ell(\tilde{\boldsymbol{\pi}}, \hat{\lambda}) - \eta H(\tilde{\boldsymbol{\pi}})])}_{(III)} + \underbrace{(\ell(\tilde{\boldsymbol{\pi}}, \hat{\lambda}) - \ell(\hat{\boldsymbol{\pi}}, \hat{\lambda}))}_{(IV)}.
\end{aligned} \tag{58}$$

We now provide upper bounds for these four terms. Denote

$$\hat{k}^* = \underset{k \in [N]}{\mathrm{argmax}} \langle \hat{\pi}^k, C^k \rangle, \quad \tilde{k}^* = \underset{k \in [N]}{\mathrm{argmax}} \langle \tilde{\pi}^k, C^k \rangle. \tag{59}$$

Since (1) and (2) are equivalent, we have the following for the term (I):

$$\begin{aligned}
(I) &= \sum_k [\bar{\lambda}(\hat{\boldsymbol{\pi}})]_k \langle \hat{\pi}^k, C^k \rangle - \sum_k [\bar{\lambda}(\tilde{\boldsymbol{\pi}})]_k \langle \tilde{\pi}^k, C^k \rangle = \langle \hat{\pi}^{\hat{k}^*}, C^{\hat{k}^*} \rangle - \langle \tilde{\pi}^{\tilde{k}^*}, C^{\tilde{k}^*} \rangle \\
&\leq \langle \hat{\pi}^{\hat{k}^*}, C^{\hat{k}^*} \rangle - \langle \tilde{\pi}^{\hat{k}^*}, C^{\hat{k}^*} \rangle \leq \|\hat{\pi}^{\hat{k}^*} - \tilde{\pi}^{\hat{k}^*}\|_1 \|C^k\|_\infty \leq c_\infty \sum_k \|\hat{\pi}^k - \tilde{\pi}^k\|_1 \\
&\leq 2c_\infty \sum_k (\|r(\tilde{\pi}^k) - a^k\|_1 + \|c(\tilde{\pi}^k) - b^k\|_1) = 2c_\infty \|c^{T-1} - b\|_1,
\end{aligned} \tag{60}$$

where the first inequality follows from the definition of $\tilde{k}^*$ in (59), the fourth inequality is from Lemma 7, and the last equality follows from (15) and (17).

For the term (II), recall that $H(\boldsymbol{\pi}) = -\sum_{k,i,j} \pi_{i,j}^k (\log \pi_{i,j}^k - 1)$ and $\tilde{\pi}^k = \exp\left(\frac{f^T \mathbf{1}^\top + \mathbf{1}(g^{T-1})^\top - \lambda_k^{T-1} C^k}{\eta}\right)$ due to (35), and define $u^{T-1} = \frac{\max_j g_j^{T-1} + \min_j g_j^{T-1}}{2}$. We

have

$$
\begin{aligned}
(II) &= \sum_k \bar{\lambda}(\tilde{\boldsymbol{\pi}})_k \langle \tilde{\pi}^k, C^k \rangle + \eta \sum_{k,i,j} \tilde{\pi}_{i,j}^k \left( \frac{f_i^T + g_j^{T-1} - \lambda_k^{T-1} C_{ij}^k}{\eta} - 1 \right) - F^* \\
&= \sum_k (\bar{\lambda}(\tilde{\boldsymbol{\pi}})_k - \hat{\lambda}_k) \langle \tilde{\pi}^k, C^k \rangle + \sum_{k,i,j} \tilde{\pi}_{i,j}^k \left( f_i^T + g_j^{T-1} - \eta \right) - F^* \\
&= \left\langle \bar{\lambda}(\tilde{\boldsymbol{\pi}}) - \hat{\lambda}, \nabla_\lambda F(f^T, g^{T-1}, \lambda^{T-1}) \right\rangle + \left\langle f^T, a \right\rangle + \left\langle g^{T-1}, c^{T-1} \right\rangle - \eta \sum_{i,j,k} \tilde{\pi}_{i,j}^k - F^* \\
&= \left\langle \bar{\lambda}(\tilde{\boldsymbol{\pi}}) - \hat{\lambda}, \nabla_\lambda F(f^T, g^{T-1}, \lambda^{T-1}) \right\rangle + \left\langle f^T, a \right\rangle + \left\langle g^{T-1}, c^{T-1} \right\rangle - \log \left( \sum_k \|\tilde{\pi}^k\|_1 \right) - \eta - F^* \\
&= \left\langle \bar{\lambda}(\tilde{\boldsymbol{\pi}}) - \hat{\lambda}, \nabla_\lambda F(f^T, g^{T-1}, \lambda^{T-1}) \right\rangle + \left\langle g^{T-1}, c^{T-1} - b \right\rangle + F(f^T, g^{T-1}, \lambda^{T-1}) - F^* \\
&\leq \left\langle \bar{\lambda}(\tilde{\boldsymbol{\pi}}) - \hat{\lambda}, \nabla_\lambda F(f^T, g^{T-1}, \lambda^{T-1}) \right\rangle + \left\langle g^{T-1}, c^{T-1} - b \right\rangle \\
&\leq c_\infty \|c^{T-1} - b\|_1 + 3c_\infty^2 \|\lambda^T - \lambda^{T-1}\|_2 / \eta + \left\langle g^{T-1} - u^{T-1} \mathbf{1}, c^{T-1} - b \right\rangle \\
&\leq c_\infty \|c^{T-1} - b\|_1 + 3c_\infty^2 \|\lambda^T - \lambda^{T-1}\|_2 / \eta + \|g^{T-1} - u^{T-1} \mathbf{1}\|_\infty \|c^{T-1} - b\|_1 \\
&\leq (3c_\infty/2 - \eta\iota/2) \|c^{T-1} - b\|_1 + 3c_\infty^2 \|\lambda^T - \lambda^{T-1}\|_2 / \eta,
\end{aligned}
\tag{61}
$$

where the third equality uses (35), (33c) and (15), the second inequality follows from Lemma 12 by setting $\lambda = \bar{\lambda}(\tilde{\boldsymbol{\pi}})$ and $t = T - 1$, and the last inequality uses Lemma 9.

For the term (III), we have

$$
\begin{aligned}
(III) &\leq \left| \sum_k \hat{\lambda}_k \langle \tilde{\pi}^k, C^k \rangle + \eta \sum_{i,j} \tilde{\pi}_{i,j}^k \left( \frac{f_i^T + g_j^{T-1} - \lambda_k^{T-1} C^k}{\eta} - 1 \right) - F^* \right| \\
&= \left| \langle g^{T-1}, c^{T-1} - b \rangle + F(f^T, g^{T-1}, \lambda^{T-1}) - F^* \right| \\
&\leq (c_\infty/2 - \eta\iota/2) \|c^{T-1} - b\|_1 + |F(f^T, g^{T-1}, \lambda^{T-1}) - F^*|,
\end{aligned}
\tag{62}
$$

where the last inequality follows from Lemma 9.

Finally, for the term (IV), we have

$$
\begin{aligned}
(IV) &= \sum_k \langle \tilde{\pi}^k - \hat{\pi}^k, \hat{\lambda}_k C^k \rangle \leq \sum_k \|\tilde{\pi}^k - \hat{\pi}^k\|_1 \|C^k\|_\infty \\
&\leq 2c_\infty \sum_k (\|r(\tilde{\pi}^k) - a^k\|_1 + \|c(\tilde{\pi}^k) - b^k\|_1) = 2c_\infty \|c^{T-1} - b\|_1,
\end{aligned}
\tag{63}
$$

where the first inequality uses $|\hat{\lambda}_k| \leq 1$, the second inequality uses Lemma 7 and (17). Plugging (60) - (63) into (58), and using (55), we obtain (56a).

Now we prove (56b). For ease of presentation, we denote

$$
\bar{\boldsymbol{\pi}}(\lambda) := \underset{\boldsymbol{\pi} \in \Pi_{a,b}^N}{\operatorname{argmin}} \ell(\boldsymbol{\pi}, \lambda).
\tag{64}
$$

We also denote $\tilde{b} = c^{T-1} = \sum_k c(\tilde{\pi}^k)$ and $\pi'^k = \operatorname{Round}(\bar{\pi}(\hat{\lambda})^k, \tilde{a}^k, \tilde{b}^k)$, where

$$
(\tilde{a}^k, \tilde{b}^k)_{k \in [N]} := \operatorname{Margins}(\bar{\boldsymbol{\pi}}(\hat{\lambda}), a, \tilde{b}),
$$

as defined in (16). From (18) we know that

$$
\sum_k \left\| c\left( (\bar{\pi}(\hat{\lambda}))^k \right) - \tilde{b}^k \right\|_1 = \left\| \sum_k c\left( (\bar{\pi}(\hat{\lambda}))^k \right) - \sum_k \tilde{b}^k \right\|_1 = \|b - \tilde{b}\|_1 = \|b - c^{T-1}\|_1,
\tag{65}
$$

where the second equality is due to $\bar{\boldsymbol{\pi}}(\hat{\lambda}) \in \Pi_{a,b}^N$ and thus $c(\sum_k (\bar{\boldsymbol{\pi}}(\hat{\lambda}))^k) = b$, and the fact that $\sum_k \tilde{b}^k = \tilde{b}$ due to Property (ii) of the Margins procedure in Section 2.1. By the Sinkhorn's theorem

Sinkhorn (1967), $\tilde{\boldsymbol{\pi}}$ is the unique optimal solution of $\min_{\boldsymbol{\pi} \in \Pi^N_{\tilde{a}, \tilde{b}}} \ell_\eta(\boldsymbol{\pi}, \hat{\lambda})$. Therefore

$$\sum_k \hat{\lambda}_k \langle \tilde{\pi}^k, C^k \rangle - \eta H(\tilde{\boldsymbol{\pi}}) \leq \sum_k \hat{\lambda}_k \langle \pi'^k, C^k \rangle - \eta H(\boldsymbol{\pi}'). \tag{66}$$

Now, note that the left hand side of (56b) can be arranged into three parts:

$$\ell(\hat{\boldsymbol{\pi}}, \hat{\lambda}) - \ell(\bar{\boldsymbol{\pi}}(\hat{\lambda}), \hat{\lambda})$$
$$= \underbrace{\left( \sum_k \hat{\lambda}_k \langle \hat{\pi}^k, C^k \rangle - \sum_k \hat{\lambda}_k \langle \tilde{\pi}^k, C^k \rangle \right)}_{(V)} + \underbrace{\left( \sum_k \hat{\lambda}_k \langle \tilde{\pi}^k, C^k \rangle - \sum_k \hat{\lambda}_k \langle \pi'^k, C^k \rangle \right)}_{(VI)} \tag{67}$$
$$+ \underbrace{\left( \sum_k \hat{\lambda}_k \langle \pi'^k, C^k \rangle - \sum_k \hat{\lambda}_k \langle (\bar{\pi}(\hat{\lambda}))^k, C^k \rangle \right)}_{(VII)}.$$

We now upper bound these three terms. First note that the term (V) is the same as the term (IV) and thus has the same upper bound in (63). Since $0 \leq H(\boldsymbol{\pi}) \leq \log(n^2 N) + 1$, from (66) we have that

$$(VI) = \sum_k \hat{\lambda}_k \langle \tilde{\pi}^k, C^k \rangle - \sum_k \hat{\lambda}_k \langle \pi'^k, C^k \rangle \leq \eta \left| H(\tilde{\boldsymbol{\pi}}) - H(\boldsymbol{\pi}') \right| \leq \frac{1}{3}\epsilon, \tag{68}$$

where the last step uses the definition of $\eta$ in (20).

For the term (VII), we have

$$(VII) = \sum_k \hat{\lambda}_k \langle \pi'^k, C^k \rangle - \sum_k \hat{\lambda}_k \langle (\bar{\pi}(\hat{\lambda}))^k, C^k \rangle \leq \sum_k \|\pi'^k - (\bar{\pi}(\hat{\lambda}))^k\|_1 \|C^k\|_\infty$$
$$\leq 2c_\infty \sum_k (\|r((\bar{\pi}(\hat{\lambda}))^k) - \tilde{a}^k\|_1 + \|c((\bar{\pi}(\hat{\lambda}))^k) - \tilde{b}^k\|_1) = 2c_\infty \|c^{T-1} - b\|_1, \tag{69}$$

where the second inequality follows from Lemma 7, the second equality uses (65) and the fact that $r((\bar{\pi}(\hat{\lambda}))^k) = \tilde{a}^k$ due to the property of the Margins procedure in (16).

Finally, plugging (63) (note (V)=(IV)), (68) and (69) into (67), and using (55a) and noting $\iota < 0$, we obtain (56b). This completes the proof.

$\square$

### B.3 MAIN RESULT

**Theorem 16** *Define $\epsilon' = \epsilon/(6c_\infty - \eta\iota)$, and set $T$ as*

$$T = 5 + \frac{36}{\eta\sqrt{\gamma_0}\epsilon'} + \frac{648c_\infty^2}{\eta\epsilon} + \frac{28}{\eta\gamma_0\epsilon} = O\left(c_\infty^2 \epsilon^{-2}\right), \tag{70}$$

*where $\gamma_0$ is defined in Lemma 14 and we know $\gamma_0 = O(c_\infty^{-2})$. The output pair of Algorithm 1 is an $\epsilon$-optimal solution of the EOT problem* (2).

*Proof.* According to Lemma 15, we only need to show that (55) holds after $T$ iterations as defined in (70). To guarantee (55a) and (55b), we follow the ideas of Dvurechensky *et al.* Dvurechensky et al. (2018) and construct a switching process. We first reduce $\tilde{F}$ from $\tilde{F}(f^0, g^0, \lambda^0)$ to a constant $s$ by running $t_1$ steps. In this process, Lemma 14 indicates

$$t_1 \leq 1 + \frac{4}{\eta\gamma_0 s} - \frac{4}{\eta\gamma_0 \tilde{F}(f^0, g^0, \lambda^0)}. \tag{71}$$

Secondly, starting from $s$, we continue running the algorithm, and assume that there are $t_2$ iterations in which (55a) fails. By (42b) we have

$$t_2 \leq 1 + \frac{72s}{\eta\epsilon'^2}.$$

Therefore, we know that the total iteration number that (55a) fails is upper bounded by

$$T_1 = t_1 + t_2 \leq 2 + \frac{72s}{\eta\epsilon'^2} + \frac{4}{\eta\gamma_0 s} - \frac{4}{\eta\gamma_0 \tilde{F}(f^0, g^0, \lambda^0)}$$

iterations. By choosing $s = \epsilon'/(6\sqrt{\gamma_0})$, we know that

$$T_1 \leq \begin{cases} 2 + \frac{12}{\eta\sqrt{\gamma_0}\epsilon'} + \frac{24}{\eta\sqrt{\gamma_0}\epsilon'} - \frac{4}{\eta\gamma_0\tilde{F}(f^0, g^0, \lambda^0)} \leq 2 + \frac{36}{\eta\sqrt{\gamma_0}\epsilon'} & \text{if } \tilde{F}(f^0, g^0, \lambda^0) \geq \frac{\epsilon'}{6\sqrt{\gamma_0}} \\ 2 + \frac{12}{\eta\sqrt{\gamma_0}\epsilon'} + \frac{24}{\eta\sqrt{\gamma_0}\epsilon'} - \frac{4}{\eta\gamma_0\tilde{F}(f^0, g^0, \lambda^0)} \leq 2 + \frac{12}{\eta\sqrt{\gamma_0}\epsilon'} & \text{otherwise.} \end{cases}$$

Therefore, we have $T_1 \leq 2 + \frac{36}{\eta\sqrt{\gamma_0}\epsilon'}$. Similarly, starting from $s$, the number of iterations that (55b) fails can be bounded by

$$t_3 \leq 1 + \frac{648sc_\infty^2}{\eta\epsilon^2},$$

where we apply (42b). By choosing $s = \epsilon$, we know that the total iteration number that (55b) fails is upper bounded by

$$T_2 = t_1 + t_3 \leq 2 + \frac{648c_\infty^2}{\eta\epsilon} + \frac{4}{\eta\gamma_0\epsilon} - \frac{4}{\eta\gamma_0\tilde{F}(f^0, g^0, \lambda^0)}$$

iterations. Finally, by letting $s = \epsilon/6$ in (71), we know that

$$\tilde{F}(f^{T_3-1}, g^{T_3-1}, \lambda^{T_3-1}) \leq \epsilon/6$$

after

$$T_3 = 1 + \frac{24}{\eta\gamma_0\epsilon}$$

iterations. From (42a) we know that after $T_3$ iterations, we have

$$\tilde{F}(f^{T_3}, g^{T_3-1}, \lambda^{T_3-1}) \leq \tilde{F}(f^{T_3-1}, g^{T_3-1}, \lambda^{T_3-1}) \leq \epsilon/6,$$

i.e., (55c) holds. Combining the above discussions, we know that after $T = T_1 + T_2 + T_3 + 1$ iterations, there must exist at least one iteration such that (55) holds, and thus the output of PAM is an $\epsilon$-optimal solution to the original EOT problem (2).

$\square$

## C    PROOF OF THEOREM 22

The following simple fact is useful for our analysis later.

$$\|y^{t+1} - \lambda^t\|_2 = \|\text{Proj}_{\Delta^N}\left(\lambda^t + (1-\theta)(\lambda^t - \lambda^{t-1})\right) - \text{Proj}_{\Delta^N}\left(\lambda^t\right)\|_2 \leq (1-\theta)\|\lambda^t - \lambda^{t-1}\|_2, \tag{72}$$

where the equality follows from the definition of $y^{t+1}$ in (26c), and the inequality is due to the non-expansiveness of the projection operator.

The following lemma shows that the Hamiltonian $E(f^t, g^t, \lambda^t, \lambda^{t-1})$ is monotonically increasing when updating $\lambda$ in Algorithm (3).

**Lemma 17** *[Sufficient increase in $\lambda$] Let $\{f^t, g^t, y^t, \lambda^t\}$ be generated by PAME (Algorithm 3). The following inequality holds:*

$$E(f^{t+1}, g^{t+1}, \lambda^{t+1}, \lambda^t) - E(f^{t+1}, g^{t+1}, \lambda^t, \lambda^{t-1}) \geq \frac{2\theta - \theta^2}{2\tau}\|\lambda^t - \lambda^{t-1}\|_2^2 + \frac{1}{4\tau}\|\lambda^{t+1} - y^{t+1}\|_2^2. \tag{73}$$

*Note that since $\theta \in (0, 1)$, the right hand side of (73) is always nonnegative.*

*Proof.* From the optimality condition of (26d) we know that, there exists $h(\lambda^{t+1}) \in \partial\mathbb{I}_{\Delta^N}(\lambda^{t+1})$ such that

$$\nabla_\lambda F(f^{t+1}, g^{t+1}, y^{t+1}) - \frac{1}{\tau}(\lambda^{t+1} - y^{t+1}) - h(\lambda^{t+1}) = 0. \tag{74}$$

By the convexity of the indicator function $\mathbb{I}_{\Delta^N}(\lambda^{t+1})$, we have

$$\langle y^{t+1} - \lambda^{t+1}, h(\lambda^{t+1})\rangle \le 0, \quad \langle \lambda^t - \lambda^{t+1}, h(\lambda^{t+1})\rangle \le 0. \tag{75}$$

Moreover, we have the following inequality:

$$\begin{aligned}
\|\lambda^{t+1} - \lambda^t\|_2^2 &= \|\lambda^{t+1} - y^{t+1} + y^{t+1} - \lambda^t\|_2^2 \\
&= \|y^{t+1} - \lambda^t\|_2^2 + 2\langle \lambda^{t+1} - y^{t+1}, y^{t+1} - \lambda^t\rangle + \|\lambda^{t+1} - y^{t+1}\|_2^2 \\
&\le (1-\theta)^2\|\lambda^t - \lambda^{t-1}\|_2^2 + 2\langle \lambda^{t+1} - y^{t+1}, y^{t+1} - \lambda^t\rangle + \|\lambda^{t+1} - y^{t+1}\|_2^2,
\end{aligned} \tag{76}$$

where the inequality is from (72).

We then have the following inequality:

$$\begin{aligned}
&F(f^{t+1}, g^{t+1}, \lambda^t) - F(f^{t+1}, g^{t+1}, \lambda^{t+1}) \\
&\le \left(F(f^{t+1}, g^{t+1}, y^{t+1}) + \langle \nabla_\lambda F(f^{t+1}, g^{t+1}, y^{t+1}), \lambda^t - y^{t+1}\rangle\right) - \\
&\quad \left(F(f^{t+1}, g^{t+1}, y^{t+1}) + \langle \nabla_\lambda F(f^{t+1}, g^{t+1}, y^{t+1}), \lambda^{t+1} - y^{t+1}\rangle - c_\infty^2\|\lambda^{t+1} - y^{t+1}\|_2^2/(2\eta)\right) \\
&= \langle \nabla_\lambda F(f^{t+1}, g^{t+1}, y^{t+1}), \lambda^t - \lambda^{t+1}\rangle + c_\infty^2\|\lambda^{t+1} - y^{t+1}\|_2^2/(2\eta) \\
&\le \langle \nabla_\lambda F(f^{t+1}, g^{t+1}, y^{t+1}) - h(\lambda^{t+1}), \lambda^t - \lambda^{t+1}\rangle + c_\infty^2\|\lambda^{t+1} - y^{t+1}\|_2^2/(2\eta) \\
&= \frac{1}{\tau}\langle \lambda^{t+1} - y^{t+1}, \lambda^t - \lambda^{t+1}\rangle + \frac{1}{4\tau}\|\lambda^{t+1} - y^{t+1}\|^2 \\
&= \frac{1}{\tau}\langle \lambda^{t+1} - y^{t+1}, \lambda^t - y^{t+1} + y^{t+1} - \lambda^{t+1}\rangle + \frac{1}{4\tau}\|\lambda^{t+1} - y^{t+1}\|^2 \\
&= -\frac{1}{\tau}\langle \lambda^{t+1} - y^{t+1}, y^{t+1} - \lambda^t\rangle - \frac{3}{4\tau}\|\lambda^{t+1} - y^{t+1}\|^2,
\end{aligned} \tag{77}$$

where the first inequality is from the concavity of $F$ with respect to $\lambda$ and (37), the second inequality is due to (75), the second equality is due to (74). Combining (76) and (77) leads to

$$\begin{aligned}
E(f^{t+1}, g^{t+1}, \lambda^{t+1}, \lambda^t) &= F(f^{t+1}, g^{t+1}, \lambda^{t+1}) - \frac{1}{2\tau}\|\lambda^{t+1} - \lambda^t\|_2^2 \\
&\ge F(f^{t+1}, g^{t+1}, \lambda^t) + \frac{1}{\tau}\langle \lambda^{t+1} - y^{t+1}, y^{t+1} - \lambda^t\rangle + \frac{3}{4\tau}\|\lambda^{t+1} - y^{t+1}\|^2 \\
&\quad - \frac{(1-\theta)^2}{2\tau}\|\lambda^t - \lambda^{t-1}\|_2^2 - \frac{1}{\tau}\langle \lambda^{t+1} - y^{t+1}, y^{t+1} - \lambda^t\rangle - \frac{1}{2\tau}\|\lambda^{t+1} - y^{t+1}\|_2^2 \\
&= F(f^{t+1}, g^{t+1}, \lambda^t) - \frac{1}{2\tau}\|\lambda^t - \lambda^{t-1}\|_2^2 + \frac{2\theta - \theta^2}{2\tau}\|\lambda^t - \lambda^{t-1}\|_2^2 + \frac{1}{4\tau}\|\lambda^{t+1} - y^{t+1}\|^2 \\
&= E(f^{t+1}, g^{t+1}, \lambda^t, \lambda^{t-1}) + \frac{2\theta - \theta^2}{2\tau}\|\lambda^t - \lambda^{t-1}\|_2^2 + \frac{1}{4\tau}\|\lambda^{t+1} - y^{t+1}\|^2,
\end{aligned}$$

which completes the proof. $\qquad\square$

Now we define the following function $\tilde{E}$, and later we will prove that $\tilde{E}(f^t, g^t, \lambda^t, \lambda^{t-1})$ can be upper bounded by $O(1/t)$.

$$\tilde{E}(f, g, \lambda^1, \lambda^2) = F(f^*, g^*, \lambda^*) - E(f, g, \lambda^1, \lambda^2).$$

The next lemma is useful for obtaining the upper bound for $\tilde{E}(f^t, g^t, \lambda^t, \lambda^{t-1})$. Moreover, it is noted that $\tilde{E}(f, g, \lambda^1, \lambda^2) \ge 0, \forall f, g, \lambda^1, \lambda^2$, and $\tilde{E}(f, g, \lambda, \lambda) = \tilde{F}(f, g, \lambda), \forall f, g, \lambda$.

**Lemma 18** *Let $\{f^t, g^t, y^t, \lambda^t\}$ be generated by PAME (Algorithm 3). For any $\lambda \in \Delta^N$, the following inequality holds*

$$\langle \lambda - \lambda^t, \nabla_\lambda F(f^{t+1}, g^t, \lambda^t)\rangle \le c_\infty\|c^t - b\|_1 + 7c_\infty^2\|\lambda^{t+1} - y^{t+1}\|_2/\eta + 5(1-\theta)c_\infty^2\|\lambda^t - \lambda^{t-1}\|_2/\eta. \tag{78}$$

*Proof.* From the optimality condition of (26d), we have the following inequality:

$$\left\langle \lambda - \lambda^{t+1}, \frac{1}{\tau}(\lambda^{t+1} - y^{t+1}) - \nabla_\lambda F(f^{t+1}, g^{t+1}, y^{t+1})\right\rangle \ge 0, \quad \forall \lambda \in \Delta^N. \tag{79}$$

The left hand side of (78) can be rearranged to three terms.

$$\langle \lambda - \lambda^t, \nabla_\lambda F(f^{t+1}, g^t, \lambda^t) \rangle$$
$$= \underbrace{\langle \lambda^t - \lambda, -\nabla_\lambda F(f^{t+1}, g^{t+1}, y^{t+1}) \rangle}_{(I)} + \underbrace{\langle \lambda^t - \lambda, \nabla_\lambda F(f^{t+1}, g^{t+1}, y^{t+1}) - \nabla_\lambda F(f^{t+1}, g^{t+1}, \lambda^t) \rangle}_{(II)}$$
$$+ \underbrace{\langle \lambda^t - \lambda, \nabla_\lambda F(f^{t+1}, g^{t+1}, \lambda^t) - \nabla_\lambda F(f^{t+1}, g^t, \lambda^t) \rangle}_{(III)}.$$

$$(80)$$

We now bound these three terms one by one. To bound the term (I), we first note that from (33c) and (15), we have

$$\|\nabla_\lambda F(f^{t+1}, g^{t+1}, \lambda^t)\|_2 \le c_\infty \le c_\infty^2/\eta, \tag{81}$$

where the second inequality is due to the definition of $\eta$ (28). Now we can bound the term (I) as follows:

$$(I) = \langle \lambda^t - \lambda^{t+1}, -\nabla_\lambda F(f^{t+1}, g^{t+1}, \lambda^t) \rangle + \langle \lambda^t - \lambda^{t+1}, \nabla_\lambda F(f^{t+1}, g^{t+1}, \lambda^t) - \nabla_\lambda F(f^{t+1}, g^{t+1}, y^{t+1}) \rangle$$
$$+ \langle \lambda^{t+1} - \lambda, -\nabla_\lambda F(f^{t+1}, g^{t+1}, y^{t+1}) \rangle$$
$$\le \left\| \lambda^t - \lambda^{t+1} \right\|_2 \cdot \left\| \nabla_\lambda F(f^{t+1}, g^{t+1}, \lambda^t) \right\|_2 + \frac{c_\infty^2}{\eta} \left\| \lambda^t - \lambda^{t+1} \right\|_2 \cdot \left\| \lambda^t - y^{t+1} \right\|_2$$
$$+ \frac{1}{\tau} \left\| \lambda^{t+1} - \lambda \right\|_2 \cdot \left\| \lambda^{t+1} - y^{t+1} \right\|_2$$
$$\le 3c_\infty^2 \|\lambda^t - \lambda^{t+1}\|_2/\eta + 4c_\infty^2 \|\lambda^{t+1} - y^{t+1}\|_2/\eta,$$

$$(82)$$

where the first inequality uses Lemma 8 and (79), the second inequality uses (81) and the facts that $\left\| \lambda^t - y^{t+1} \right\|_2 \le 2$ and $\left\| \lambda^t - \lambda \right\|_2 \le 2$.

For the term (II), Lemma 8 yields:

$$(II) \le 2 \left\| \nabla_\lambda F(f^{t+1}, g^{t+1}, y^{t+1}) - \nabla_\lambda F(f^{t+1}, g^{t+1}, \lambda^t) \right\|_2 \le 2c_\infty^2 \left\| y^{t+1} - \lambda^t \right\|_2/\eta. \tag{83}$$

For the term (III), it can be bounded as:

$$(III) = \sum_{k=1}^N (\lambda_k^t - \lambda_k) \cdot \langle \pi^k(f^{t+1}, g^{t+1}, \lambda^t) - \pi^k(f^{t+1}, g^t, \lambda^t), C^k \rangle$$
$$\le \sum_{k=1}^N \|\pi^k(f^{t+1}, g^{t+1}, \lambda^t) - \pi^k(f^{t+1}, g^t, \lambda^t)\|_1 \|C^k\|_\infty \le c_\infty \|c^t - b\|_1,$$

$$(84)$$

where the last inequality is due to Lemma (10). Plugging (82) - (84) into (80) and applying the triangle inequality, we obtain

$$\langle \lambda - \lambda^t, \nabla_\lambda F(f^{t+1}, g^t, \lambda^t) \rangle \le c_\infty \|c^t - b\|_1 + 7c_\infty^2 \|\lambda^{t+1} - y^{t+1}\|_2/\eta + 5c_\infty^2 \|y^{t+1} - \lambda^t\|_2/\eta,$$

which immediately implies (78) by noting (72).

$\square$

**Lemma 19** Let $(f^t, g^t, y^t, \lambda^t)$ be generated by PAME (Algorithm 3). The following inequality holds:

$$\tilde{E}(f^{t+1}, g^t, \lambda^t, \lambda^{t-1}) \le (2c_\infty - \eta\iota)\|c^t - b\|_1 + 7c_\infty^2 \|\lambda^{t+1} - y^{t+1}\|_2/\eta + (7 - 5\theta)c_\infty^2 \|\lambda^t - \lambda^{t-1}\|_2/\eta.$$

*Proof.* Since $F(f, g, \lambda)$ is a concave function, we have

$$F(f^*, g^*, \lambda^*) \le F(f^{t+1}, g^t, \lambda^t) + \langle \nabla F(f^{t+1}, g^t, \lambda^t), (f^*, g^*, \lambda^*) - (f^{t+1}, g^t, \lambda^t) \rangle,$$

which implies that

$$\tilde{F}(f^{t+1}, g^t, \lambda^t) \le \langle g^t - g^*, c^t - b \rangle + \langle \lambda^t - \lambda^*, -\nabla_\lambda F(f^{t+1}, g^t, \lambda^t) \rangle$$
$$\le (c_\infty - \eta\iota)\|c^t - b\|_1 + c_\infty \|c^t - b\|_1 + 7c_\infty^2 \|\lambda^{t+1} - y^{t+1}\|_2/\eta \tag{85}$$
$$+ 5(1 - \theta)c_\infty^2 \|\lambda^t - \lambda^{t-1}\|_2/\eta.$$

where in the first inequality we have used (35), and the second inequality follows from (50) and setting $\lambda = \lambda^*$ in (78). From (85) we immediately get

$$\tilde{E}(f^{t+1}, g^t, \lambda^t, \lambda^{t-1}) = \tilde{F}(f^{t+1}, g^t, \lambda^t) + \frac{1}{2\tau}\|\lambda^t - \lambda^{t-1}\|_2^2$$

$$\leq c_\infty^2\|\lambda^t - \lambda^{t-1}\|_2^2/\eta + (2c_\infty - \eta\iota)\|c^t - b\|_1 + 7c_\infty^2\|\lambda^{t+1} - y^{t+1}\|_2/\eta + 5(1-\theta)c_\infty^2\|\lambda^t - \lambda^{t-1}\|_2/\eta$$

$$\leq 2c_\infty^2\|\lambda^t - \lambda^{t-1}\|_2/\eta + (2c_\infty - \eta\iota)\|c^t - b\|_1 + 7c_\infty^2\|\lambda^{t+1} - y^{t+1}\|_2/\eta + 5(1-\theta)c_\infty^2\|\lambda^t - \lambda^{t-1}\|_2/\eta,$$

where the second inequality is due to $\|\lambda^t - \lambda^{t-1}\|_2 \leq 2$. This completes the proof. $\qquad\square$

The following lemma bounds $\tilde{E}(f^{t+1}, g^{t+1}, \lambda^{t+1}, \lambda^t)$ by $O(1/t)$.

**Lemma 20** *Let $\{f^t, g^t, y^t\lambda^t\}$ be generated by PAME (Algorithm 3). The following inequality holds:*

$$\tilde{E}(f^{t+1}, g^{t+1}, \lambda^{t+1}, \lambda^t) \leq \frac{6/(\eta\gamma_1)}{t + 1 + 6/(\eta\gamma_1\tilde{F}(f^0, g^0, \lambda^0))}, \forall t \geq 0,$$

*where we assume $\lambda^{-1} = \lambda^0$, and*

$$\gamma_1 = \min\left\{\frac{1}{(2c_\infty - \eta\iota)^2}, \frac{2(2\theta - \theta^2)}{(7 - 5\theta)^2 c_\infty^2}, \frac{1}{49c_\infty^2}\right\} \tag{86}$$

*is a constant.*

*Proof.* Combining (42b) and Lemma 17, we have

$$E(f^{t+1}, g^{t+1}, \lambda^{t+1}, \lambda^t) - E(f^{t+1}, g^t, \lambda^t, \lambda^{t-1})$$

$$= \left(E(f^{t+1}, g^{t+1}, \lambda^{t+1}, \lambda^t) - E(f^{t+1}, g^{t+1}, \lambda^t, \lambda^{t-1})\right) + \left(F(f^{t+1}, g^{t+1}, \lambda^t) - F(f^{t+1}, g^t, \lambda^t)\right)$$

$$\geq \frac{\eta}{2}\|c^t - b\|_1^2 + \frac{2\theta - \theta^2}{2\tau}\|\lambda^t - \lambda^{t-1}\|_2^2 + \frac{1}{4\tau}\|\lambda^{t+1} - y^{t+1}\|_2^2,$$

which implies that

$$\tilde{E}(f^{t+1}, g^{t+1}, \lambda^{t+1}, \lambda^t) - \tilde{E}(f^{t+1}, g^t, \lambda^t, \lambda^{t-1})$$

$$\leq -\frac{\eta}{2}\|c^t - b\|_1^2 - (2\theta - \theta^2)\frac{c_\infty^2}{\eta}\|\lambda^t - \lambda^{t-1}\|_2^2 - \frac{c_\infty^2}{2\eta}\|\lambda^{t+1} - y^{t+1}\|_2^2$$

$$\leq -\frac{\eta}{2}\gamma_1\left[\left((2c_\infty - \eta\iota)\|c^t - b\|_1\right)^2 + \left((7 - 5\theta)c_\infty^2\|\lambda^t - \lambda^{t-1}\|_2/\eta\right)^2 + \left(7c_\infty^2\|\lambda^{t+1} - y^{t+1}\|_2/\eta\right)^2\right]$$

$$\leq -\frac{\eta}{6}\gamma_1\left[(2c_\infty - \eta\iota)\|c^t - b\|_1 + (7 - 5\theta)c_\infty^2\|\lambda^t - \lambda^{t-1}\|_2/\eta + 7c_\infty^2\|\lambda^{t+1} - y^{t+1}\|_2/\eta\right]^2$$

$$\leq -\frac{\eta}{6}\gamma_1\tilde{E}(f^{t+1}, g^t, \lambda^t, \lambda^{t-1})^2, \tag{87}$$

where the last inequality applies Lemma 19. We then divide both sides of (87) by $\tilde{E}(f^{t+1}, g^{t+1}, \lambda^{t+1}, \lambda^t) \cdot \tilde{E}(f^{t+1}, g^t, \lambda^t, \lambda^{t-1})$, and we obtain

$$\frac{1}{\tilde{E}(f^{t+1}, g^{t+1}, \lambda^{t+1}, \lambda^t)} \geq \frac{1}{\tilde{E}(f^{t+1}, g^t, \lambda^t, \lambda^{t-1})} + \frac{\eta}{6}\gamma_1 \cdot \frac{\tilde{E}(f^{t+1}, g^t, \lambda^t, \lambda^{t-1})}{\tilde{E}(f^{t+1}, g^{t+1}, \lambda^{t+1}, \lambda^t)}$$

$$\geq \frac{1}{\tilde{E}(f^{t+1}, g^t, \lambda^t, \lambda^{t-1})} + \frac{\eta}{6}\gamma_1 \geq \frac{1}{\tilde{E}(f^t, g^t, \lambda^t, \lambda^{t-1})} + \frac{\eta}{6}\gamma_1, \tag{88}$$

where the second inequality holds because (87) implies that $\tilde{E}(f^{t+1}, g^t, \lambda^t, \lambda^{t-1}) \geq \tilde{E}(f^{t+1}, g^{t+1}, \lambda^{t+1}, \lambda^t)$, and the last inequality follows from (42a). Summing (88) from 0 to $t$ leads to

$$\frac{1}{\tilde{E}(f^{t+1}, g^{t+1}, \lambda^{t+1}, \lambda^t)} \geq \frac{1}{\tilde{E}(f^0, g^0, \lambda^0, \lambda^{-1})} + \frac{\eta(t+1)}{6}\gamma_1 = \frac{1}{\tilde{F}(f^0, g^0, \lambda^0)} + \frac{\eta(t+1)}{6}\gamma_1,$$

which immediately leads to the desired result. □

Similar to Lemma 15, the following lemma provides some sufficient conditions for the PAME algorithm to return an $\epsilon$-optimal solution to the original EOT problem (2).

**Lemma 21** *Assume PAME terminates at the $T$-iteration, i.e.,*

$$\|c^{T-1} - b\|_1 \leq \epsilon/(6(6c_\infty - \eta\iota), \tag{89a}$$

$$\|\lambda^{T-1} - \lambda^{T-2}\|_2 \leq \eta\epsilon/(60(1-\theta)c_\infty^2), \tag{89b}$$

$$\|\lambda^T - y^T\|_2 \leq \eta\epsilon/(42c_\infty^2), \tag{89c}$$

$$\tilde{F}(f^T, g^{T-1}, \lambda^{T-1}) \leq \epsilon/6. \tag{89d}$$

*Then the output $(\hat{\pi}, \hat{\lambda})$ of PAME (Algorithm 3), i.e., $\hat{\pi}^k = Round(\pi^k(f^T, g^{T-1}, \lambda^{T-1}), a^k, b^k), \forall k \in [N], \hat{\lambda} = \lambda^{T-1}$, is an $\epsilon$-optimal solution of the original EOT problem* (2).

*Proof.* The proof is essentially the same as that of Lemma 15. More specifically, we again need to show that the output of PAME $(\hat{\pi}, \hat{\lambda})$ satisfies (56). The proof of (56b) is exactly the same as the proof of Lemma 15. The proof of (56a) only requires to develop a new bound for

$$\left\langle \bar{\lambda}(\tilde{\pi}) - \hat{\lambda}, \nabla_\lambda F(f^T, g^{T-1}, \lambda^{T-1}) \right\rangle \tag{90}$$

that is used in (61). Other parts are again exactly the same as the ones in Lemma 15. The new bound of (90) can be obtained by applying Lemma 18 with $\lambda = \bar{\lambda}(\tilde{\pi})$ and $t = T - 1$, which yields

$$
\begin{aligned}
&\langle \bar{\lambda}(\tilde{\pi}) - \hat{\lambda}, \nabla_\lambda F(f^T, g^{T-1}, \lambda^{T-1}) \rangle \\
&\leq c_\infty\|c^{T-1} - b\|_1 + 5(1-\theta)c_\infty^2\|\lambda^{T-1} - \lambda^{T-2}\|_2/\eta + 7c_\infty^2\|\lambda^T - y^T\|_2/\eta.
\end{aligned}
\tag{91}
$$

By combining (91) with (60)-(63), we can bound the left hand side of (56a) by

$$
\begin{aligned}
&\ell\left(\hat{\pi}, \bar{\lambda}(\hat{\pi})\right) - \ell(\hat{\pi}, \hat{\lambda}) \\
&\leq (6c_\infty - \eta\iota)\left\|c^{T-1} - b\right\|_1 + 5(1-\theta)c_\infty^2\left\|\lambda^{T-1} - \lambda^{T-2}\right\|_2/\eta + 7c_\infty^2\|\lambda^T - y^T\|_2/\eta \\
&\quad + \left|F(f^T, g^{T-1}, \lambda^{T-1}) - F^*\right| \\
&\leq \left(\frac{1}{6} + \frac{1}{12} + \frac{1}{12} + \frac{1}{6}\right)\epsilon = \frac{1}{2}\epsilon,
\end{aligned}
\tag{92}
$$

where in the last inequality we have used all the sufficient conditions (89a)-(89d). □

**Theorem 22** *Define $\epsilon' = \epsilon/(6c_\infty - \eta\iota)$, and set $T$ to be*

$$T = 8 + \frac{48}{\eta\sqrt{\gamma_1}\epsilon'} + \frac{\left(3600(1-\theta)^2 + 882\right)c_\infty^2 s}{\eta\epsilon^2} + \frac{48}{\eta\gamma_1\epsilon} = O\left(c_\infty^2\epsilon^{-2}\right), \tag{93}$$

*where $\gamma_1$ is defined in (86) and we know $\gamma_1 = O(c_\infty^{-2})$. At least one of the iterations in Algorithm 3, after rounding, is an $\epsilon$-saddle point of the EOT problem* (2).

*Proof.* According to Lemma 21, we only need to show that (89) holds after $T$ iterations as defined in (93).

We follow the same idea as the proof of Theorem 16. First we reduce $\tilde{E}(f^{t+1}, g^{t+1}, \lambda^{t+1}, \lambda^t)$ from $\tilde{E}(f^0, g^0, \lambda^0, \lambda^{-1}) = \tilde{F}(f^0, g^0, \lambda^0)$ to a constant $s$ by running $t_1$ steps. By Lemma 20, we have

$$t_1 \leq 1 + \frac{6}{\eta\gamma_1 s} - \frac{6}{\eta\gamma_1\tilde{F}(f^0, g^0, \lambda^0)}. \tag{94}$$

Secondly, starting from $s$, we continue running the algorithm, and assume that there are $t_2$ iteration in which (89a) fails. By (42b) we have

$$t_2 \leq 1 + \frac{72s}{\eta\epsilon'^2}.$$

Therefore, we know that the total iteration number that (89a) fails can be upper bounded by

$$T_1 = t_1 + t_2 \leq 2 + \frac{72s}{\eta\epsilon'^2} + \frac{6}{\eta\gamma_1 s} - \frac{6}{\eta\gamma_1 \tilde{F}(f^0, g^0, \lambda^0)}$$

iterations. By choosing $s = \frac{\epsilon'}{6\sqrt{\gamma_1}}$, we know that

$$T_1 \leq \begin{cases} 2 + \frac{12}{\eta\sqrt{\gamma_1}\epsilon'} + \frac{36}{\eta\sqrt{\gamma_1}\epsilon'} - \frac{6}{\eta\gamma_1 \tilde{F}(f^0, g^0, \lambda^0)} \leq 2 + \frac{48}{\eta\sqrt{\gamma_1}\epsilon'} & \text{if } \tilde{F}(f^0, g^0, \lambda^0) \geq \frac{\epsilon'}{6\sqrt{\gamma_1}}, \\ 2 + \frac{12}{\eta\sqrt{\gamma_1}\epsilon'} + \frac{36}{\eta\sqrt{\gamma_1}\epsilon'} - \frac{6}{\eta\gamma_1 \tilde{F}(f^0, g^0, \lambda^0)} \leq 2 + \frac{12}{\eta\sqrt{\gamma_1}\epsilon'} & \text{otherwise.} \end{cases}$$

Therefore, we have $T_1 \leq 2 + \frac{48}{\eta\sqrt{\gamma_1}\epsilon'}$. Similarly, from Lemma 17 we know that, starting from $s$, the number of iterations that (89b) and (89c) fail can be respectively bounded by

$$t_3 \leq 1 + \frac{3600(1-\theta)^2 c_\infty^2 s}{\eta\epsilon^2(2\theta - \theta^2)}, \quad \text{and} \quad t_4 \leq 1 + \frac{3528 c_\infty^2 s}{\eta\epsilon^2}.$$

By choosing $s = \epsilon$, we have the total iteration numbers that (89b) and (89c) fail can be respectively bounded by

$$T_2 = t_1 + t_3 \leq 2 + \frac{3600(1-\theta)^2 c_\infty^2}{\eta\epsilon(2\theta - \theta^2)} + \frac{6}{\eta\gamma_1\epsilon} - \frac{6}{\eta\gamma_1 \tilde{F}(f^0, g^0, \lambda^0)} \leq 2 + \frac{3600(1-\theta)^2 c_\infty^2}{\eta\epsilon(2\theta - \theta^2)} + \frac{6}{\eta\gamma_1\epsilon}$$

and

$$T_3 = t_1 + t_4 \leq 2 + \frac{3528 c_\infty^2}{\eta\epsilon} + \frac{6}{\eta\gamma_1\epsilon} - \frac{6}{\eta\gamma_1 \tilde{F}(f^0, g^0, \lambda^0)} \leq 2 + \frac{3528 c_\infty^2}{\eta\epsilon} + \frac{6}{\eta\gamma_1\epsilon}.$$

Finally, by letting $s = \epsilon/6$ in (94), we know that

$$\tilde{E}(f^{T_4-1}, g^{T_4-1}, \lambda^{T_4-1}, \lambda^{T_4-2}) \leq \epsilon/6 \tag{95}$$

after

$$T_4 = 1 + \frac{36}{\eta\gamma_1\epsilon}$$

iterations. From (95) we know that

$$\tilde{F}(f^{T_4-1}, g^{T_4-1}, \lambda^{T_4-1}) \leq \epsilon/6,$$

which implies that (89d) holds with $T = T_4$ by noting (42a).

Combining the above discussions, we know that after $T = T_1 + T_2 + T_3 + T_4 + 1$ iterations, there must exist at least one iteration such that the sufficient condition (89) holds, and thus the output of PAME is an $\epsilon$-optimal solution to the original EOT problem (2). $\qquad\square$

# D ADDITIONAL NUMERICAL RESULTS

## D.1 GAUSSIAN DISTRIBUTION

Figure 2 shows the optimal couplings obtained from the standard OT and EOT of two Gaussian distributions under three different metrics: the Euclidean cost ($\|\cdot\|_2$), the square Euclidean cost ($\|\cdot\|_2^2$) and the $L_1^{1.5}$ norm ($\|\cdot\|_1^{1.5}$) respectively. We set $n = 4, \eta = 0.05$ and generate samples independently according to (31). For the EOT problem, we consider three agents with cost matrices computed by the three metrics mentioned above. Note that the entropy regularized models lead to a dense transportation plan and Figure 2 only plots the couplings with a probability larger than $10^{-3}$. We see that all the agents have the same total cost in the EOT model, and as expected, the cost is smaller than the other three OT costs obtained by using the same metric. The sub figures in the first row imply that if we split the workload to three parts evenly, then the three agents will each have costs 5.935/3, 2.158/3 and 5.030/3, which is not fair because they have different costs. But the EOT model can indeed guarantee the fairness.

We further compare the computational time for Gaussian distributions. We generate the data as in Section 5 and set the parameters as $\eta = 0.5, \tau = 5\eta/c_\infty^2$. We stop all the algorithms when the EOT error (32) is less than $10^{-4}$. Tables 1 and 2 show the CPU time (in seconds) for different $(n, N)$ pairs. The reported computational time is averaged over 5 runs. In Table 1, the APGA algorithm fails to reach an error of $10^{-4}$ in 500000 iterations when $n = 100$ and $n = 500$. We conclude that the APGA algorithm converges much slower than PAM and PAME algorithms. The PAME algorithm performs the best among all three algorithms.

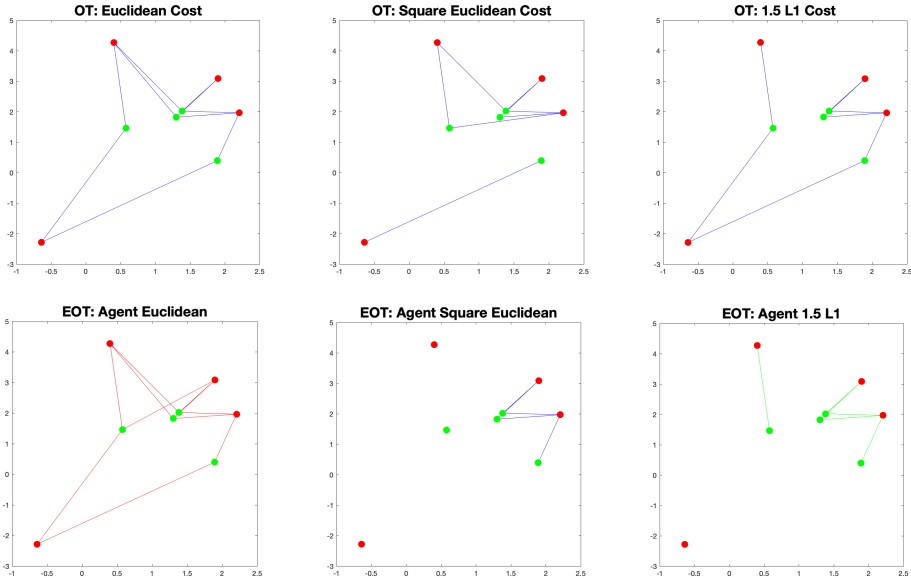

Figure 2: Optimal couplings of standard OT (first row) and EOT (second row). OT Square Euclidean Cost: 5.935; OT Euclidean Cost: 2.158; OT $L_1^{1.5}$ Cost: 5.030; EOT Cost: 0.906.

Table 1: CPU time (in seconds) comparison for Gaussian Distributions. Fixed $N = 3$.

| Algorithms | $n = 10$ | $n = 20$ | $n = 50$ | $n = 100$ | $n = 500$ |
|---|---|---|---|---|---|
| PAM | 0.038283 | 0.096039 | 0.177209 | 1.038785 | 1.768560 |
| PAME | 0.025210 | 0.065552 | 0.091593 | 0.564618 | 1.340104 |
| APGA | 1.125673 | 12.364775 | 106.768840 | - | - |

## D.2 Numerical Results on Fragmented Hypercube Dataset

In this section, we compare the performance of PAME with PAM and APGA (25) Scetbon et al. (2021) on the fragmented hypercube dataset.

**Fragmented Hypercube:** We now consider transferring mass between a uniform distribution over a hypercube $\mu = \mathcal{U}([-1, 1]^d)$ and a distribution $\nu$ obtained by a pushforward $\nu = T_\sharp \mu$ defined by $T(x) = x + 2\text{sign}(x) \odot \left( \sum_{m=1}^{m^*} e_m \right)$. Here $\text{sign}(\cdot)$ is taken elementwise, $m^* \in [d]$ and $e_i$, $i \in [d]$ is the canonical basis of $\mathbb{R}^d$. In our experiments, we set $d = 10, m^* = 2$ and sample two base support sets $\{x_i^{base}\}_{i \in [n]}, \{y_j^{base}\}_{j \in [n]}$ independently from $\mu, \nu$. To obtain the cost matrix for one agent, we first add Gaussian noise sampled from $\mathcal{N}(0, 1)$ to the base support sets to get $\{x_i^{noisy}\}_{i \in [n]}, \{y_j^{noisy}\}_{j \in [n]}$ and compute the cost using the noisy support sets. For instance, for the $k$-th agent, we have $(x_i^{noisy})^k = x_i^{base} + \mathcal{N}(0, 1)$, $(y_j^{noisy})^k = y_j^{base} + \mathcal{N}(0, 1)$ and $C_{i,j}^k = \|(x_i^{noisy})^k - (y_j^{noisy})^k\|_2^2$.

Figures 3 plots the EOT error versus the CPU time for Fragmented Hypercube dataset. We run PAM for 20000 iterations to get an approximate optimal $\ell^*$ and run all algorithms for 2000 iterations for different parameter settings. In all cases, the PAME and PAM perform significantly better than APGA, and PAME also shows significant improvement over PAM.

We then compare the computational time for Fragmented Hypercube dataset. We set the parameters as $\eta = 0.2, \tau = 5\eta/c_\infty^2$. We stop all the algorithms when the EOT error (32) is less than $10^{-4}$. Tables 3 and 4 show the CPU time (averaged over 5 runs) for different $(n, N)$ pairs. We see that the PAME algorithm still performs the best among all three algorithms. Note that in Table 3 the APGA algorithm fails to reach an error of $10^{-4}$ in 500000 iterations when $n = 50$, $n = 100$ and $n = 500$.

Table 2: CPU time (in seconds) comparison for Gaussian Distributions. Fixed $n = 50$.

| Algorithms | $N = 2$ | $N = 3$ | $N = 5$ | $N = 10$ | $N = 20$ |
|---|---|---|---|---|---|
| PAM | 0.180343 | 0.177209 | 1.021598 | 0.719909 | 0.903429 |
| PAME | 0.105775 | 0.091593 | 0.560785 | 0.385495 | 0.618381 |
| APGA | 70.959300 | 106.768840 | 169.213121 | 178.637889 | 212.697866 |

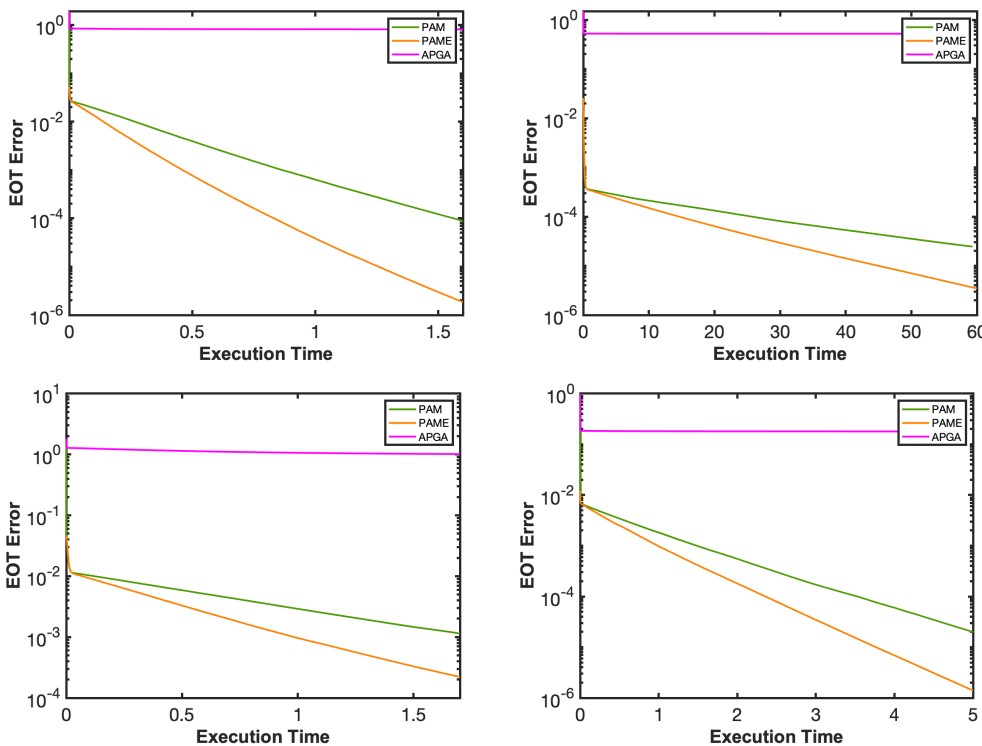

Figure 3: CPU time comparison between PAM, PAME and APGA algorithms on the Fragmented Hypercube dataset. **Upper Left**: $N = 5, n = 100, \eta = 0.2$, **Upper Right**: $N = 5, n = 500, \eta = 0.2$, **Bottom Left**: $N = 5, n = 100, \eta = 0.1$, **Bottom Right**: $N = 10, n = 100, \eta = 0.2$.

Table 3: CPU time (in seconds) comparison for Fragmented Hypercube. Fixed $N = 3$.

| Algorithms | $n = 10$ | $n = 20$ | $n = 50$ | $n = 100$ | $n = 500$ |
|---|---|---|---|---|---|
| PAM | 0.165165 | 0.101363 | 0.177209 | 1.154193 | 3.840804 |
| PAME | 0.112527 | 0.068529 | 0.091593 | 0.588553 | 2.007653 |
| APGA | 13.771911 | 22.017804 | - | - | - |

Table 4: CPU time (in seconds) comparison for Fragmented Hypercube. Fixed $n = 20$.

| Algorithms | $N = 2$ | $N = 3$ | $N = 5$ | $N = 10$ | $N = 20$ |
|---|---|---|---|---|---|
| PAM | 0.003180 | 0.101363 | 0.172696 | 0.253926 | 0.231646 |
| PAME | 0.040302 | 0.068529 | 0.110080 | 0.150513 | 0.129156 |
| APGA | 1.007166 | 22.017804 | 13.801154 | 6.304324 | 3.749495 |

