# OpenReview forum: "On the Convergence of Projected Alternating Maximization for Equitable and Optimal Transport"
_ICLR.cc/2022/Conference — ICLR 2022 Submitted_

### Official Review · Reviewer_9vP7 · 2021-10-28

**Correctness:** 4
**Technical Novelty And Significance:** 2
**Empirical Novelty And Significance:** 2
**Recommendation:** 5
**Confidence:** 4

**Main Review:**


This work solves a somewhat narrowly scoped problem of EOT: equitable optimal transport, together with entropy regularization, as an intriguing novel formulation. However, what is unclear is why the entropy regularization is necessary, or in what sense it helps. I agree that strong concavity helps with rate analysis, especially as it is necessary to obtain improved rates from Nesterov acceleration.

 However, for strongly concave problems, the best achievable rate is linear/exponential, and often invokes mirror-prox style updates. These aspects are not treated anywhere in the manuscript, so I wonder whether additional improvements are possible. Some more rigorous/granular discussion of how the main convergence results address these points is missing.

On the practical side, a few different use cases of EOT are discussed in the introduction  such as cake-cutting, multi-type resource allocation, internet minimal transportation time, etc. However, none of these are credibly addressed in the experiments, which implicitly asks the reviewer to take it on faith that EOT is applicable to these domains. This seems like a major drawback of the manuscript. The authors would do well to properly develop a single application that corroborates the theory. Put another way, why are the Gaussian distributions and the fragmented hypercube dataset representative of practical problems?

In summary, this paper develops some interesting theoretical insights about optimal transport with entropy regularization, but the innovations mostly borrow from existing results that are standard in numerical optimization, machine learning with regularization, and while the rate analysis is presented for a novel context, it does not itself employ any novel techniques or technical innovations. On top of that, the experiments are extremely simple and do not convincingly demonstrate that EOT is useful in a problem that arises in practice.


**Summary Of The Paper:**

This paper considers the problem of approximate optimal transport, where population probability measures are replaced by discrete PMFs at collections of samples, and there is an additional "equitable" constraint that seeks to ``fairly" distribute the probability mass. This admits a formulation as a linear program and may be solved efficiently through its dual. However, recovering the primal solution is nontrivial, and there exists no rate analysis to-date. This work develops a stochastic dual ascent method to solve an entropy-regularized version of this LP, where entropy regularization is introduced to ensure strong concavity, as well as avoid a notion of over-fitting. The crux of this method is an alternating block coordinate ascent algorithm, which is then combined with an extrapolation/Nesterov acceleration scheme to accelerate convergence. Numerical experiments demonstrate the practical utility of the proposed approach.


**Summary Of The Review:**

This work is well-written but not particularly technically innovative, and is missing a thoroughly convincing use-case.

---

> ### Author Response · Authors · 2021-11-18
> **Response to reviewer's comments**
>
> 1. Response to the comment on the discussions related to the convergence results:
>
>  The entropy regularization was first introduced into OT problem by [Cuturi, 2013] and is now widely used in the OT community. By adding an entropy regularizer, the primal problem becomes strongly convex and the dual problem is unconstrained and is suitable for alternating maximization. This leads to the Sinkhorn's algorithm which has low per-iteration complexity and thus is scalable. The PAM algorithm proposed by [Scetbon et al. 2021] used the same idea for the EOT problem. We agree that for strongly convex/concave problems, the best achievable rate is linear. But Sinkhorn's algorithm solves the dual problem, which is not strongly concave. The reason why people don't solve the strongly convex primal problem (i.e., the entropic regularized OT problem) is that it involves constraints that don't admit easy projection. We believe these are the main reasons why the literature has been focusing on the dual problem of OT. As for the PAM algorithm, we have tried to develop an accelerated algorithm (i.e., PAME) based on Nesterov's technique, however we are not able to analytically prove that PAME has an improved complexity bound at this moment yet. The APGA proposed in [Scetbon et al. 2021] in fact has better complexity than PAM and PAME. However, as demonstrated in [Scetbon et al. 2021] and in our paper, APGA  performs worse than PAM. We believe the reason is that APGA takes gradient step for the variables $f$ and $g$, while PAM exactly minimizes the subproblems corresponding to these two variables. It is the exact minimization step that led to the improvement. Developing a provably better algorithm is definitely important and interesting, and we will work on it in the future. These discussions are added to the revised paper (see the texts marked in blue color on page 3 and the Remark 6).
>
> 2. Response to the comment on the experiment:
>
>  It is true that we only conducted numerical experiments on toy examples. The main purpose was to provide some supports and justifications to our theoretical results in this paper. The Gaussian distributions and fragmented hypercube example are only some examples where the OT can be easily computed analytically. These distributions have been widely used in the OT literature to demonstrate the main ideas of the algorithms. We are not claiming that they are representative for more practical problems. We agree that it is very important to implement our algorithms for solving real application of EOT, and we will work on it in a future work.
>
> We added more numerical results in Appendix D. In particular, we added Tables 1-4 in Appendix D. These tables report the CPU time comparisons among PAM, PAME and APGA for varying $N$ and $n$. These results further confirmed that both PAM and PAME are more efficient than APGA, and PAME performs the best among the three algorithms.

---

### Official Review · Reviewer_1p7i · 2021-11-01

**Correctness:** 3
**Technical Novelty And Significance:** 3
**Empirical Novelty And Significance:** 1
**Recommendation:** 6
**Confidence:** 3

**Main Review:**

Strengths:
(a) The paper is well-written as easy to follow.
(b) The paper provides missing convergence guarantees for the interesting problem of EOT.
(c) The authors further provide a novel rounding scheme plus a variant of EOT.

Weaknesses:
(a) My main concern is about the novelty and the level of contribution of the paper, which is mostly providing analysis of an existing algorithm for an existing probem.
(b) The experimental section is quite limited and its presentation is rushed: it only considers synthetic data. I also have difficulty interpreting the results reported in Fig. 2.

**Summary Of The Paper:**

The authors focus on the equitable and optimal transport (EOT) problem. In particular, EOT is a linear programming (LP) problem, which is expensive to solve in practice. The existing work suggests adding an entropy term to the objective and solving the resulting problem in the dual domain via an algorithm named Projected Alternating Maximization algorithm (PAM). However, there are two main issues with PAM: that is its convergence is unknown and it only provides a solution in the dual domain. The authors precisely address these issues by providing a convergence rate for PAM and proposing a novel method to compute a feasible primal solution from the dual solution. Furthermore, they propose a variant of PAM, i.e., PAM with Extrapolation (PAME), which appears to perform better in practice.

**Summary Of The Review:**

Overall, the paper provides some novel theoretical results for the PAM algorithm that is used for solving EOT. The paper is well-written and its theoretical results are strong. However, my main concern is the scope of the contributions. I am personally not familiar with EOT applications, but if other reviewers believe that EOT is a sufficiently interesting problem, I will be willing to vote for acceptance.

---

> ### Author Response · Authors · 2021-11-18
> **Response to reviewer's comments**
>
> 1. Response to the comment on novelty and contribution:
>
> We believe that EOT is an important problem with promising potential in practical applications. See our response to reviewer rMWc. Thus it is very important to establish the convergence guarantee for the PAM algorithm for solving EOT.
>
>
> 2. Response to the comment on experimental section and Fig. 2:
>
> We added more numerical results in Appendix D. In particular, we added Tables 1-4 in Appendix D. These tables report the CPU time comparisons among PAM, PAME and APGA for varying $N$ and $n$. These results further confirmed that both PAM and PAME are more efficient than APGA, and PAME performs the best among the three algorithms. We agree that it is very important to implement our algorithms for solving real application of EOT, and we will work on it in a future work.
>
> We moved Fig. 2 to the Appendix D. The sub figures in the first row of Figure 2 are the OT costs when there is one single agent. The sub figures in the second row of Figure 2 are the EOT costs of each of the agent when there are three agents. These results have two implications. First, the EOT cost is lower than the OT cost. This is intuitive because it is expected that when there are multiple agents, the cost of each agent is lower than the cost of one single agent doing the whole job. Second, EOT is fair. The sub figures in the second row show that the three agents have the same cost 0.906. The sub figures in the first row imply that if we split the workload to three parts evenly, then the three agents will each have costs 5.935/3, 2.158/3 and 5.030/3, respectively. This is not fair because they have different costs. So the EOT model can indeed guarantee the fairness.

---

### Official Review · Reviewer_M8bm · 2021-11-04

**Correctness:** 4
**Technical Novelty And Significance:** 4
**Empirical Novelty And Significance:** 4
**Recommendation:** 8
**Confidence:** 4

**Main Review:**

Strengths
1. The authors provide a novel, useful and thorough convergence analysis for the PAM algorithm for the EOT problem. The convergence analysis is facilitated by the introduction of a novel procedure for extracting the primal solution.
2. The paper proposes an acceleration procedure, and provides a convergence proof. The authors cannot theoretically establish a faster rate, but the numerical experiments indicate a substantial improvement.
Weaknesses
1. There is limited discussion of the main result. The authors do draw connections with (hybrid) BCD and BCGD methods, although this is more to explain why existing convergence analysis results cannot be applied.
2. The experimental section is limited. It focuses on a single synthetic example. While it is certainly sufficient considering that the primary contribution of the paper is the theoretical development, more extensive experiments (and associated analysis) would strengthen the paper.
Comments:
The paper provides a useful and novel theoretical result for the EOT problem. The paper is very well-written and provides clear explanations. The proofs are laid out clearly with a good development of lemmas so that the overall structure can be easily followed.
I think the paper could be improved by some further discussion of the main result. For example, there is an observation in the contributions section that the rate is the same as that of Sinkhorn’s algorithm for computing the Wasserstein distance. I would appreciate some further discussion along these lines, and also some comments regarding the c_\infty term.
Given that the paper is primarily theoretical, the experimental results are sufficient, but they do not represent a systematic, thorough examination of the algorithm, focusing on a single toy example (or two, if we include the supplementary), and restricting attention to the case of uniform distributions.
There is not a strong tie between the theoretical results and the experiments. There is a single figure showing the EOT error with respect to execution time for different configurations of N, n, and \eta, but it is challenging to draw too many conclusions from these figures. No attempt is made to connect the depicted results to the theoretical result.
Figure 2 provides a comparison of the optimal couplings of OT versus EOT for different metrics. The figure could be explained more clearly. The figure provides a nice illustration of the difference between EOT and OT, although I am not convinced that such an illustration is particularly relevant to the main contribution of this paper. The conclusion  drawn from this figure is not particularly insightful – “the [EOT] cost is smaller than the other three OT costs obtained by using the same metric”.


**Summary Of The Paper:**

The paper addresses the equitable and optimal transport (EOT) problem. One approach to this problem involves perturbation via the addition of an entropy regularization. For this formulation, researchers have proposed a projected alternating maximization algorithm (PAM). The authors of this paper provide a convergence analysis of the PAM algorithm. They also introduce a novel rounding procedure to construct the primal solution of the EOT problem. The paper also contributes a variant of PAM that leads to numerical performance improvements. Numerical experiments are provided for a synthetic dataset.

**Summary Of The Review:**

I have recommended “Accept”.
I think the theoretical contribution is novel and valuable. I followed the proofs and did not identify any concerns with the arguments.

The experimental component of the paper could be improved, and some additional discussion of the main results would be appreciated.

---

> ### Author Response · Authors · 2021-11-18
> **Response to reviewer's comments**
>
>
> 1. Response to the comment on the experimental section:
>
> We added more numerical results in Appendix D. In particular, we added Tables 1-4 in Appendix D. These tables report the CPU time comparisons among PAM, PAME and APGA for varying $N$ and $n$. These results further confirm that both PAM and PAME are more efficient than APGA, and PAME performs the best among the three algorithms.
>
> 2. Response to comment on the complexity and the $c_\infty$ term :
>
> Though our complexity result matches the rate of the Sinkhorn's algorithm in terms of the dependence on $\epsilon$, we argue that EOT is a more difficult problem than the entropic regularized OT, and thus our results are promising. First, EOT is a saddle-point problem while entropic regularized OT is a minimization problem. Second, the extra variable $\lambda$ in EOT requires a gradient projection step in the PAM algorithm, which introduces significant difficulty to the analysis of the convergence behavior. While for Sinkhorn's algorithm it is much easier analyze, because the dual is unconstrained. Third, since there are multiple agents in EOT, it is more difficult to design the rounding procedure to obtain the primal solution.
>
> Regarding the $c_\infty$ term. We note that the dependence of $c_\infty$ in our result and in the result of Sinkhorn's algorithm [Dvurechensky et al. 2018] are both $c_\infty^2$. We have added these discussions into the revised paper (See Remark 4).
>
> 3. Response to comment on Figure 2:
>
> We moved Fig. 2 to the Appendix D. The sub figures in the first row of Figure 2 are the OT costs when there is one single agent. The sub figures in the second row of Figure 2 are the EOT costs of each of the agent when there are three agents. These results have two implications. First, the EOT cost is lower than the OT cost. This is intuitive because it is expected that when there are multiple agents, the cost of each agent is lower than the cost of one single agent doing the whole job. Second, EOT is fair. The sub figures in the second row show that the three agents have the same cost 0.906. The sub figures in the first row imply that if we split the workload to three parts evenly, then the three agents will each have costs 5.935/3, 2.158/3 and 5.030/3, respectively. This is not fair because they have different costs. So the EOT model can indeed guarantee the fairness.

---

### Official Review · Reviewer_rMWc · 2021-11-09

**Correctness:** 4
**Technical Novelty And Significance:** 2
**Empirical Novelty And Significance:** 1
**Recommendation:** 6
**Confidence:** 4

**Main Review:**

Strengths.
- This manuscript recognizes and provides clear references for the OT and EOT problem, and clearly positions itself with respect to previous work.
- The paper is well written, apart from minor typos, and easy to follow.
- The theoretical analysis seem to be correct to the best of my knowledge. However, I did not checked every line in all proofs.

Weakness.
- This was a missed chance to really motivate why EOT is an interesting problem. There are some references, but I cannot see why would one want to use EOT.
-The numerical experiments are toy examples, of low dimension.
- Is a pity that there is no provable accelerated for the proposed accelerated method.
- Contributions are rather incremental, yet still very useful.

**Summary Of The Paper:**

This paper develops new algorithms, convergence rate analysis and additional tools for the equitable and optimal transport problem. Following the same history of OT, where we moved from the OT formulation, to the entropic regularization, to primal-dual analysis to explicit convergence analysis, this manuscript extends the idea of EOT and provides useful tools with provable performance for the practitioner. The contributions of this paper are on the complexity analysis, as well as new algorithms for the EOT problem.

**Summary Of The Review:**

Correctness: The techniques use for proving the results are rather classical in primal-dual methods.

Novelty: I gave a 2 precisely because this is a rather clear extension of the original EOT what make use of classical analysis tools to further describe the performance of the EOT algorithms. There is no new tools, but rather the use of old tools on a problem that looks very similar mathematically.

Empirical: The numeric section is very small, with toy examples of low dimensions, etc. It show it works but really does not provide any additional insights or interesting applications.

I think this paper provides sufficient results for publication.

---

> ### Author Response · Authors · 2021-11-18
> **Response to reviewer's concerns**
>
> 1. Response to the comment on the motivation of EOT:
>
> EOT is a relaxation of the fair division (fair cake-cutting) problem. The fair division problem has important applications in economics and computational choice. For example, in e-commerce, one may want to fairly divide advertisement space or broadcast time for different users based on their preferences. The fair cake-cutting problem assumes that the cake is a set of elements, and each element is an indivisible unit. Therefore, cake-cutting problem aims to find a paritition of the set. As discussed in reference [Scetbon et al. 2021], EOT is a relaxation of this problem and it assumes that there is a divisible amount of each element of the cake. In
> that case, the cake is no longer a set but a distribution and we want to divide it among the agents according to their coupled preferences. We believe that this is a very important application of EOT.
>
> 2. Response to the comment on the numerical experiments:
>
> It is true that we only conducted numerical experiments on toy examples. The main purpose was to provide some supports and justifications to the theoretical results obtained in this paper. We agree that it is very important to implement our algorithms for solving real applications of EOT, and we will work on it in a future work.
>
> 3. Response to the comment on the complexity of the accelerated method:
>
> Since we can only apply Nesterov's acceleration to the projected gradient step, the complexity of the exact maximization steps becomes the bottleneck of our final complexity. But numerically the proposed PAME performs significantly better. Deriving algorithms with a provably better complexity is an important future direction.

---

### Decision · Program_Chairs · 2022-01-20

**Decision:**

Reject

**Comment:**

The paper considers the Equitable and Optimal Transport (EOT) problem which is arises in fair division of goods and multi-resource allocation. The resulting problem is a linear program, which is polynomial-time solvable; however, the existing polynomial-time solvers either do not scale well with the dimension or are dual methods with entropic regularization for which it is unclear how to extract a primal solution. The paper shows how to extract a primal solution and also provides complexity analysis of a recently proposed projected alternating minimization method (PAM). The paper further provides a Nesterov accelerated variant of PAM.

Overall, the paper is a meaningful contribution and was considered borderline. On one side, EOT seems like an interesting problem, the paper is well-presented, and the provided complexity results are technically sound. On the other hand, the reviewers felt that the EOT problem was not motivated enough, that the techniques for proving the results were mostly standard, and that the numerical experiments were insufficient. Even though the authors provided additional numerical experiments, I did not find the responses regarding motivation (particularly in the context of ML applications) and technical novelty convincing enough. The paper could have gone in either direction, but as there was ultimately no particularly strong support from any of the reviewers, I recommend rejection. The authors are advised to carefully revise the paper and resubmit.